# Zinc finger homeobox-3 (ZFHX3) orchestrates genome-wide daily gene expression in the suprachiasmatic nucleus

Akanksha Bafna[1,2]*, Gareth Banks[1,3], Vadim Vasilyev[4], Robert Dallmann[4,5], Michael H Hastings[6], Patrick M Nolan[1]*

[1]Medical Research Council, Harwell Science Campus, Didcot, United Kingdom; [2]Nuffield Department of Clinical Neurosciences, University of Oxford, Oxfordshire, United Kingdom; [3]Nottingham Trent University, Nottingham, United Kingdom; [4]Division of Biomedical Sciences, Warwick Medical School, University of Warwick, Coventry, United Kingdom; [5]Zeeman Institute for Systems Biology and Infectious Disease Epidemiology Research, University of Warwick, Coventry, United Kingdom; [6]MRC Laboratory of Molecular Biology, Cambridge, United Kingdom

*For correspondence:
akanksha.bafna@ndcn.ox.ac.uk (AB);
pmnolan10@gmail.com (PMN)

Competing interest: The authors declare that no competing interests exist.

## eLife Assessment

This is an **important** study that generates an inventory of accessible genomic regions bound by a transcription factor ZFHX3 within the suprachiasmatic nucleus in the hypothalamus and details the impact of its depletion on daily rhythms in behaviour and gene expression patterns. Analysis using circadian phase-estimation algorithms makes the argument that gene regulatory networks are at play and changes in gene expression of a few clock genes cannot account for the observed animal behaviour. While the transcriptome analysis is **compelling**, the data on the activity of the TF in rhythmic gene expression is **solid**, and interpretations that allow for direct and/or indirect roles have been incorporated.

**Abstract** The mammalian suprachiasmatic nucleus (SCN), situated in the ventral hypothalamus, directs daily cellular and physiological rhythms across the body. The SCN clockwork is a self-sustaining transcriptional-translational feedback loop (TTFL) that in turn coordinates the expression of clock-controlled genes (CCGs) directing circadian programmes of SCN cellular activity. In the mouse, the transcription factor, ZFHX3 (zinc finger homeobox-3), is necessary for the development of the SCN and influences circadian behaviour in the adult. The molecular mechanisms by which ZFHX3 affects the SCN at transcriptomic and genomic levels are, however, poorly defined. Here, we used chromatin immunoprecipitation sequencing to map the genomic localization of ZFHX3-binding sites in SCN chromatin. To test for function, we then conducted comprehensive RNA sequencing at six distinct times-of-day to compare the SCN transcriptional profiles of control and ZFHX3-conditional null mutants. We show that the genome-wide occupancy of ZFHX3 occurs predominantly around gene transcription start sites, co-localizing with known histone modifications, and preferentially partnering with clock transcription factors (CLOCK, BMAL1) to regulate clock gene(s) transcription. Correspondingly, we show that the conditional loss of ZFHX3 in the adult has a dramatic effect on the SCN transcriptome, including changes in the levels of transcripts encoding elements of numerous neuropeptide neurotransmitter systems while attenuating the daily oscillation of the clock TF *Bmal1*. Furthermore, various TTFL genes and CCGs exhibited altered circadian expression profiles, consistent with an advanced in daily behavioural rhythms under 12 h light–12 h dark conditions. Together, these findings reveal the extensive genome-wide

regulation mediated by ZFHX3 in the central clock that orchestrates daily timekeeping in mammals.

## Introduction

Circadian (approximately 1 day) clocks are internal biological timekeepers coordinating coherent 24 h molecular, behavioural, and physiological rhythms that are synchronized with changing environmental conditions across day and night. In mammals, the circadian clock mechanism consists of cell-autonomous transcription-translation feedback loops (TTFLs) that drive rhythmic, ~24 h expression patterns of canonical clock components (*Takahashi et al., 2008*). The primary TTFL involves rhythmic transcription of period (*Per1*, *Per2*, and *Per3*) and cryptochrome genes (*Cry1* and *Cry2*). Thereafter, PER and CRY proteins form heterodimers that act on the dimeric complex of CLOCK (circadian loco-motor output cycles kaput) and BMAL1 (brain and muscle ARNT-Like 1) to repress their own transcription, which is mediated by E-box (*CACGTG*) *cis*-regulatory sequences at their gene promoters. A secondary feedback loop is driven by *Nr1d1* (*Rev-erbα*), which is also directly transcribed through binding of CLOCK:BMAL1 at E-boxes. NR1D1 feeds back to repress *Bmal1* transcription by competing with a retinoic acid-related orphan receptor (ROR) activator to bind ROR response elements (RREs) in the *Bmal1* promoter. Along with this, CLOCK:BMAL1 heterodimers also trans-activate a series of clock-controlled gene (CCGs), many of which encode transcription factors (TFs) that regulate additional downstream programmes of gene expression (*Herzog et al., 2017*; *Hogenesch et al., 2003*).

At an organismal level, mammalian circadian rhythms are directed by the suprachiasmatic nucleus (SCN) of the hypothalamus, which synchronizes robust 24 h oscillations across peripheral tissues (*Hastings et al., 2018*; *Hastings et al., 2019*; *Schibler et al., 2015*). In the SCN, the cell-autonomous TTFL molecular clocks in individual cells are synchronized through intercellular coupling to form a coherent oscillatory network (*Herzog et al., 2017*; *Maywood et al., 2021*). This intercellular coupling is maintained through several neuropeptide neurotransmitter systems, including vasoactive intestinal peptide (VIP), arginine vasopressin (AVP), and gastrin-releasing peptide (GRP) systems (*Brown et al., 2005*), while Prokinetin2 (PROK2) serves as a pace-making element in the SCN (*Morris et al., 2021*). Moreover, zinc finger homeobox-3, ZFHX3, is a key TF in the SCN. It is highly expressed in discrete populations of adult SCN cells and exerts robust control over circadian behavioural rhythms. Intriguingly, while a dominant missense mutation in *Zfhx3* (short circuit, *Zfhx3^{Sci/+}*) results in a short circadian period in mice (*Parsons et al., 2015*), a conditional null *Zfhx3* mutation results in more severe behaviour with short circadian period and arrhythmic mice (*Wilcox et al., 2017*).

Given the crucial role of the SCN in orchestrating daily timekeeping in mammals, it is imperative to understand the role of ZFHX3 in regulating the molecular circadian clock, but, although the effects of mutations in *Zfhx3* are known at the level of circadian behaviour, its molecular functions that underpin daily timekeeping are not well-characterized. Therefore, we set out to assess systematically the function of ZFHX3 by employing a multi-omics approach in the SCN. First, we analysed the genomic binding pattern of ZFHX3 by conducting ZFHX3 targeted chromatin immunoprecipitation (ChIP-seq) of SCN tissue to identify the genomic scope of ZFHX3 and its potential mechanistic roles. Second, we implemented a detailed transcriptional profiling of the SCN in control and *Zfhx3* conditional null mutants using vibratome-based microdissection (*Bafna et al., 2023b*). This allowed us to map the effect of ZFHX3 on tissue-specific expression of both rhythmic and non-rhythmic genes.

Overall, we have defined genome-wide occupancy of ZFHX3 in the SCN, showing that it is found predominantly around actively transcribed genes and is accompanied by modified histones and other SCN TFs. We noted that conditional (adult) deletion of ZFHX3 drastically modifies the SCN transcriptome by affecting genes important for circadian timekeeping, neuropeptide signalling and synaptic processes. An in-depth investigation on time-of-day dependent gene expression clearly highlighted loss of *Bmal1* rhythmic transcription along with aberrant 24 h oscillations for TTFL and non-TTFL genes. This change in rhythmic clock gene transcriptional profiles contributed towards a phase-advanced internal central clock mirroring the advance seen in the daily behavioural activity patterns of the null mutant mice. These findings reveal a pivotal and genome-wide physiological role for ZFHX3 in the maintenance of robust circadian timekeeping in the SCN and identify molecular substrates that underpin this role.

## Results

### ZFHX3 binds to active promoter sites in the SCN

Genome-wide binding of ZFHX3 was profiled in the SCN using ChIP-seq. Briefly, genomic DNA was extracted from C57BL/6J mice raised in standard 12 h/12 h light-dark conditions at two distinct time-points; *Zeitgeber* time (ZT) 03 (3 hours after lights on) and ZT15, and subjected to ChIP-seq ('Materials and methods'). The sequenced reads were aligned to the mouse genome and significant sites of ZFHX3 occupancy were assessed in the context of previously identified modified histone binding regions (*Bafna et al., 2023a*) in the SCN. We found a total of 43,236 ZFHX3-bound sites, overlapping with approximately 22% of total modified histone regions in the SCN genome. About 60% of the ZFHX3 bound sites were found near promoters (≤ 3 kb of transcription start sites [TSS]) and so would be likely to identify direct transcriptional targets of ZFHX3, whilst ca. 21% were in distal intergenic regions potentially involved in other modes of gene regulation.

When assessed in relation to modified histone binding, ZFHX3 sites fell into four categories. First, the majority of ZFHX3 peaks coincided with both H3K4me3 (trimethylation of histone H3 at lysine 4) and H3K27ac (histone 3 lysine 27 acetylation) occupancy (48.9% of total ZFHX3 peaks at ZT3, *Supplementary file 1*), and were focussed near TSS (TSS ±3 kb) (*Figure 1B*). Since such sites are likely responsible for active transcription (*Beacon et al., 2021*), the ZFHX3 occupancy in these sites highlighted probable transcriptional regulation by ZFHX3 in the SCN. Next, a subset of ZFHX3 sites (20.7%) were bound only by H3K4me3 (no H3K27ac) as illustrated at *Figure 1C*. These are indicative of poised transcriptional regulation (*Calo and Wysocka, 2013*) and were also enriched near the TSS. About 8.2% of ZFHX3 sites were bound by H3K27ac exclusively (no H3K4me3) (*Figure 1D*). These sites were commonly found at intergenic and intronic regions and suggestive of enhancer-mediated gene regulation (*Won et al., 2008*). Finally, 22% of ZFHX3 sites were found to be devoid of modified histone binding and were also concentrated at distal intergenic regions (39.04%) (*Figure 1D*). Overall, the broad occurrence of ZFHX3 alongside histone modifiers near the gene TSS and at distal intergenic regions in the SCN genome suggests its crucial role in gene regulation potentially via promoter–enhancer interaction.

Next, we assigned each ZFHX3 site to its closest transcribed gene (±3 kb) and performed functional annotation upon these genes. The functional annotation of genes which were associated with ZFHX3 and both the H3K4me3 and H3K27ac histone modifications was found to be associated with neurodegeneration and axonal guidance, synaptic processes known to be important in the SCN (glutamatergic, GABAeric, and dopaminergic systems) and various signalling pathways. Importantly, they also included pathways involved with circadian entrainment and timekeeping (*Figure 1—figure supplement 1A*), highlighting the reciprocal intersection between ZFHX3 and the SCN cell-autonomous clock. In contrast, genomic regions cooperatively bound by ZFHX3 and H3K4me3 but with no H3K27ac, or sites with ZFHX3 alone, were found to be associated with genes involved in several fundamental biological processes such as DNA replication or MAPK signalling, but notably not circadian rhythm (*Figure 1—figure supplement 1B and C*). Therefore, it is plausible that ZFHX3 binds at the TSS and regulates actively transcribing genes, particularly those involved in tissue-specific functions such as daily timekeeping, while its occupancy without histone modifiers and/or at poised state regulates genes responsible for other basal functions. Interestingly, it is worth noting that ZFHX3 occupancy did not differ appreciably between the two timepoints tested (ZT03 vs. ZT15): ZFHX3 was seen to bind consistently at the same genomic loci (*Supplementary file 1*). This indicates that other factors/mechanisms confer temporal control to these sites whilst the presence of ZFHX3 is permissive to such *cis*- and *trans*-regulatory cues.

We next examined the signature DNA binding motifs/*cis*-regulatory elements occupied by ZFHX3, either alone or with specific histone modifications (H3K4me3, H3K27ac). As anticipated, this revealed an over-representation of AT-rich motifs (*Parsons et al., 2015*). Notably, it also suggested co-occurrence of ZFHX3 with SCN developmental and maintenance TFs such as DLX (distal-less homeobox) and LHX (LIM homeobox) (*VanDunk et al., 2011*) across the tested categories (*Figure 1—figure supplement 1D*). Furthermore, ZFHX3-bound sites were also enriched for non-AT-rich motifs such as the RFX family of TFs, which have been implicated in transcriptional responses to light in the SCN (*Araki et al., 2004*). Interestingly, we noted enrichment of ARNT (bHLH) domains but only where ZFHX3 peaks were seen adjacent to H3K27ac and H3K4me3 (i.e., active promoters). This clearly points towards the possibility of shared binding of ZFHX3 with the circadian clock TFs BMAL1:CLOCK

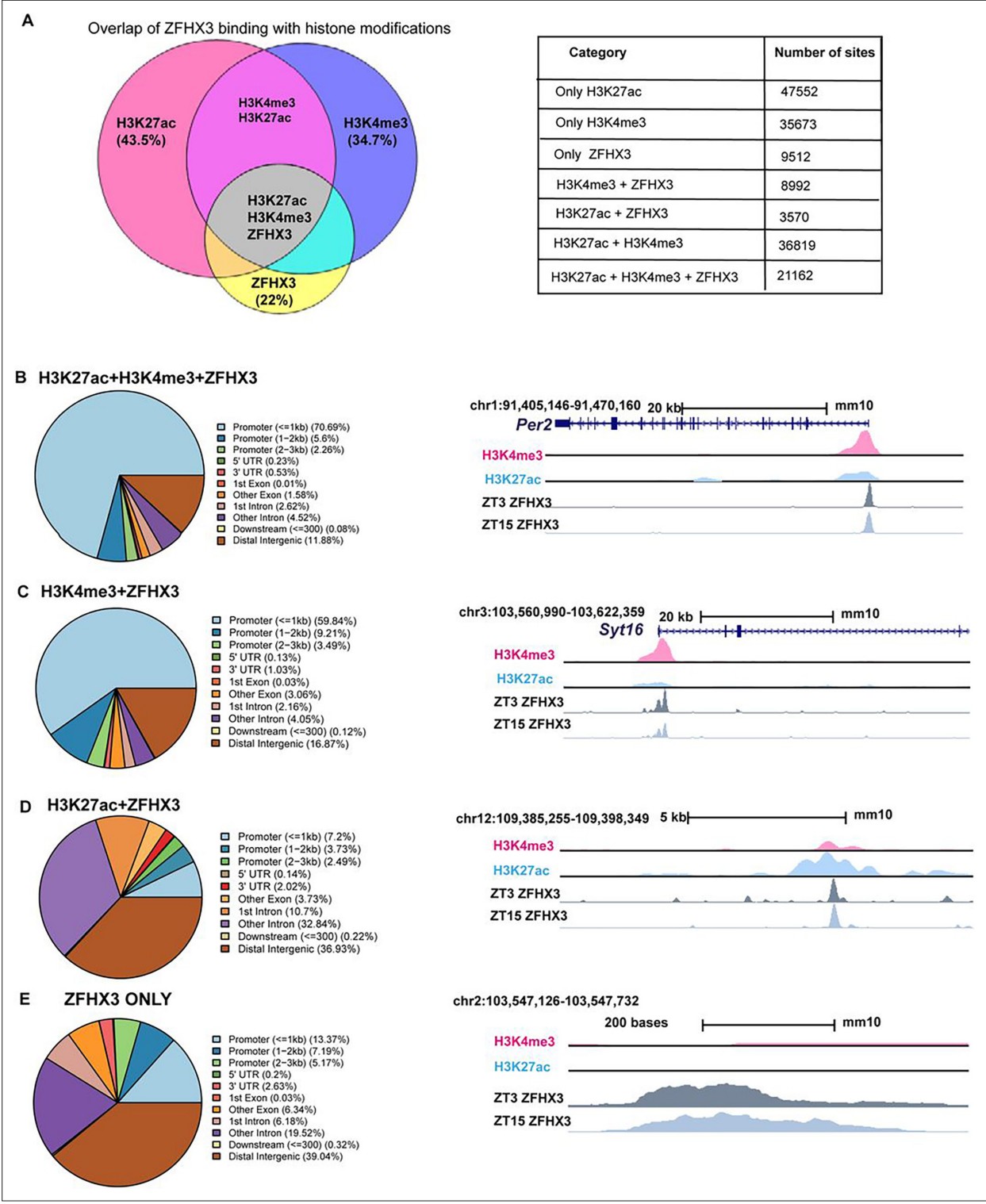

**Figure 1.** Genomic occupancy of ZFHX3 in the suprachiasmatic nucleus (SCN). (**A**) Left: Venn diagram illustrating overlap of ZFHX3 peaks with histone marks, H3K4me3 and H3K27ac; right: table showing the number of sites co-occupied by ZFHX3 and histone marks. (**B–E**) Left: genomic feature distribution of ZFHX3 peaks with histone marks (ChIPSeeker); right: UCSC Genome Browser tracks showing histone modifications H3K4me3 (pink), H3K27ac (blue), and ZFHX3 (ZT3 = dark grey and ZT15 = light grey) normalized ChIP-seq read coverage at representative examples for each category. The chromosome location and scale (mm10 genome) indicated at the top.

The online version of this article includes the following figure supplement(s) for figure 1:

**Figure supplement 1.** Gene ontology and motif analysis of ZFHX3 bound sites.

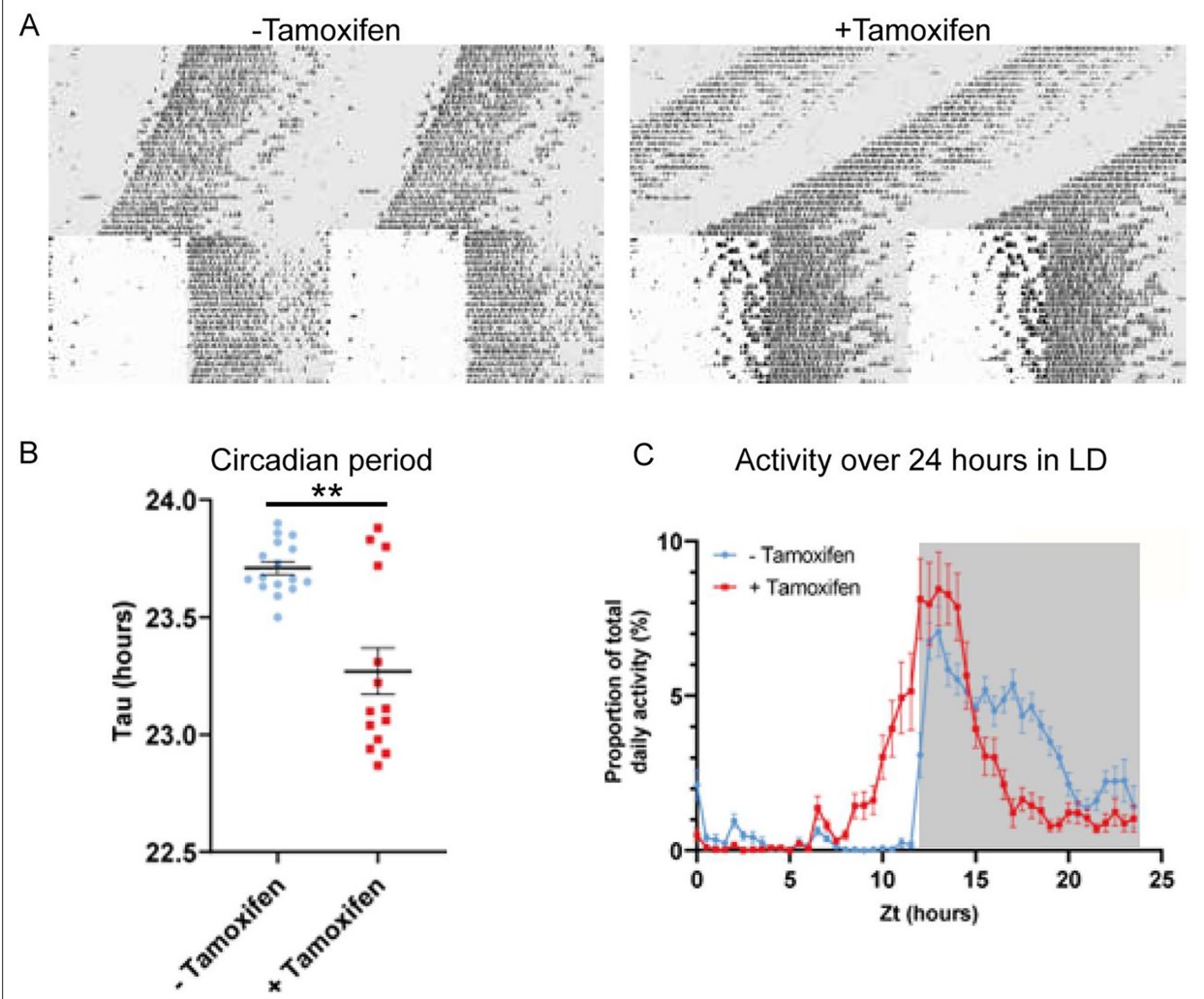

**Figure 2.** Circadian wheel running analysis of control and mutant animals. (**A**) Double-plotted actograms showing wheel running activity of −Tamoxifen and +Tamoxifen animals through 32 days of constant darkness and 24 days under a 12 h light/dark cycle. Regions in grey denote periods in darkness. (**B**) +Tamoxifen animals show a reduced circadian period. (**C**) Graph showing the distribution of activity in −Tamoxifen and +Tamoxifen animals over 24 h in a 12 h light/dark cycle. Data is taken from the final 7 days the animals were in the light/dark cycle shown in (**A**). Grey denotes the dark period of the cycle.

The online version of this article includes the following figure supplement(s) for figure 2:

**Figure supplement 1.** +Tamoxifen animals show changes in circadian wheel running parameters in conditions of constant darkness.

at E-boxes and highlights a point of interaction between the core TTFL and ZFHX3 in the regulation of circadian gene transcription in the central pacemaker.

### *Zfhx3* knock-out affects circadian behaviour and entrainment of mice

To explore the contribution of *Zfhx3* to SCN function in the adult animal, we compared circadian wheel-running in cohorts of *Zfhx3*[Flox/Flox];UBC-CreERT2 mice that had been dosed with tamoxifen (gene deletion, +Tamoxifen group) with mice that had not been dosed with tamoxifen (no deletion genetic control, -Tamoxifen group). Comparisons of gene expression between the −Tamoxifen and +Tamoxifen groups by real-time PCR confirmed loss of *Zfhx3* expression in the +Tamoxifen group. (*Figure 2— figure supplement 1A*). Behavioural analysis was performed in conditions of constant darkness and in a 12 h light/12 h dark cycle. Consistent with previous studies, this analysis showed that the free-running circadian period was significantly shortened by conditional deletion of *Zfhx3* (*Figure 2A*). Furthermore, the overall activity rhythm was less well defined with significantly reduced amplitude, increased intra-daily variability and lower inter-daily stability in the deleted mice (*Figure 2—figure*

*supplement 1B–D*). The full analysis is given in *Supplementary file 2*. The loss of ZFHX3 therefore compromised free-running circadian behaviour.

Visual inspection of the actograms whilst the animals were in light/dark cycles suggested that animals with gene deletion showed poor entrainment to the light cycle, with animals showing increased activity in the hours prior to the light to dark transition (*Figure 2A*, right). To quantify this, we analysed the distribution of wheel running activity across 24 h by expressing wheel running activity within each 30 min segment of the day as a proportion of the total daily wheel running activity for that animal (*Figure 2C*). Repeated measures ANOVA demonstrated a significant interaction between tamoxifen dosing and time of day (tamoxifen dosing × time: $F(47, 1316) = 9.013$; $p<0.0001$). Pairwise analysis demonstrated that, compared to genetic control animals, gene-deleted animals showed a significant increase in wheel running activity in the hours prior to lights off ($p<0.05$ for ZT 10.5–12.5) and a significant reduction in wheel running activity in the hours prior to lights on ($p<0.05$ for ZT 20–23). Consistent with this, analysis of the activity onset demonstrated that gene-deleted animals showed an earlier onset of activity than controls (*Figure 2—figure supplement 1E*). This phase advance under 12 h light/12 h dark is consistent with the shortening of circadian period seen under DD. Thus, ZFHX3 contributes to circadian timekeeping and entrainment in the adult SCN, independently of any developmental roles.

## ZFHX3 knock-out (KO) results in extensive effects across the SCN transcriptome

Having mapped the genomic binding sites of ZFHX3 and validated the behavioural effects of deletion of ZFHX3, we then employed total directional RNA sequencing to define the impact of *Zfhx3* deletion on the SCN transcriptome. Adult SCN tissues were collected from tamoxifen-treated *Zfhx3*$^{Flox/Flox}$-;UBC-Cre$^-$ (Cre-neg) and *Zfhx3*$^{Flox/Flox}$;UBC-Cre$^+$ mice that were not treated with tamoxifen (-Tamoxifen, genetic control), both serving as independent control groups, and tamoxifen-treated *Zfhx3*$^{Flox/Flox}$-;UBC-Cre$^+$ (+Tamoxifen) mice. that is, *Zfhx3*-deleted, at six distinct times of light/dark cycle ('Materials and methods'). As an initial analysis, we compared overall (i.e., time-independent) gene expression levels between the groups (Cre-neg vs. +Tamoxifen) to examine the global effect of *Zfhx3* deletion on the SCN transcriptome and found that almost 36% (n = 5637) of total assessed genes were affected by the acute loss of *Zfhx3* in the SCN. Of these, 2725 were downregulated and 2912 upregulated in +Tamoxifen animals compared to Cre-neg controls. (*Supplementary file 3*, *Figure 3A*). Furthermore, as expected, both the control groups (Cre-neg and -Tamoxifen) showed consistent differential gene expression levels compared to the mutant which mitigated any potential downstream effect of tamoxifen dosing per se (*Figure 3B and C*).

To better understand the relationship between ZFHX3 occupancy on the genome and changes in gene expression, we compared the results of our initial CHIP-seq with the RNA-seq described above. Of the total differentially expressed genes, 80% (n = 4543) revealed ZFHX3 occupancy at their promoter sites along with histone modifications (H3K4me3 and H3K27ac), showing a direct interaction between ZFHX3 bound TSS (±3 kb) and gene expression (p=0, OR = 6.3, Jaccard index = 0.4, signifying strong association). In contrast, only 7% of differentially expressed genes (n = 381) were solely bound by ZFHX3 at the TSS (p=0.37, OR = 1, Jaccard index = 0.1, signifying weak association), indicating the effect of ZFHX3 on gene transcription is predominantly observed when jointly occupied by histone modification at gene TSS.

We then sub-selected highly differential genes ($\log_2$FC >1, FDR <0.05) and conducted functional gene enrichment analysis. Consistent with previous studies (*Hughes et al., 2021*; *Parsons et al., 2015*; *Wilcox et al., 2021*; *Wilcox et al., 2017*), genes that were downregulated were seen to be associated with neuropeptide signalling pathways (*Vip, Grp*), circadian rhythm, neuron differentiation etc., while those that were upregulated showed functional enrichment for cell differentiation and extracellular matrix organization processes (*Figure 3D*). Interestingly, our comprehensive assessment also highlighted an effect of loss of ZFHX3 on signal transduction by affecting cholinergic receptor (*Chrna3, Chrnb4*), melatonin receptor (*Mtnr1a*), prokineticin receptor (*Prokr2*), GRP receptor (*Grpr*), and the SCN peptide *Nmu*. Therefore, it is likely that loss of ZFHX3 leads to physiological impairment by affecting a diversity of neuropeptide neurotransmitter systems in the SCN and more generally, circadian-relevant transcripts.

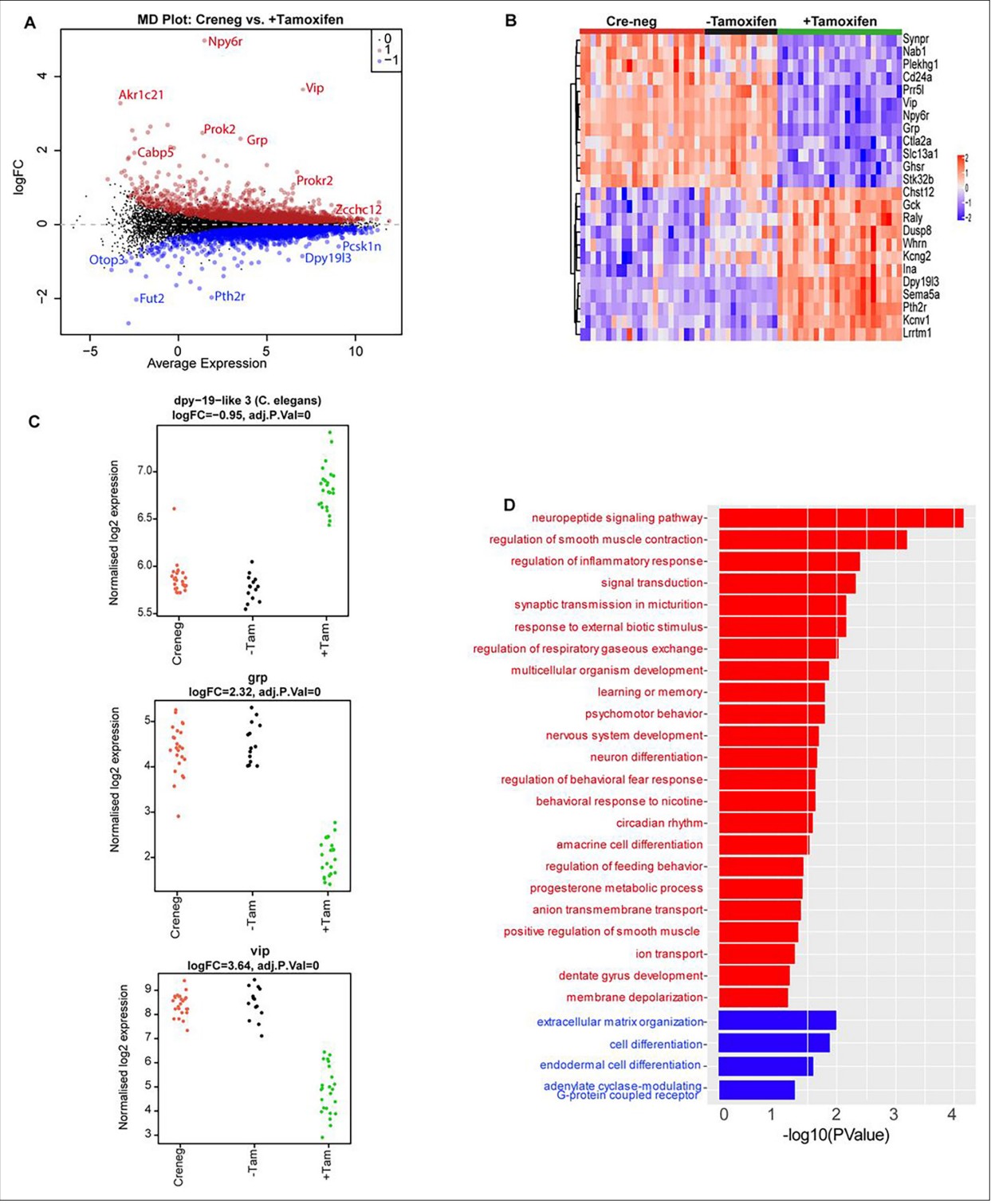

**Figure 3.** Effect of ZFHX3-KO on suprachiasmatic nucleus (SCN) transcriptome. (**A**) Mean difference plot (MD-plot) showing downregulated (red) and upregulated (blue) gene expression after the loss of ZFHX3. (**B**) Heatmap showing normalized expression of top 25 (by adj pvalue) differential genes for the compared groups. (**C**) Stripcharts of differentially expressed genes (*Dpy19l13, Grp, Vip*). (**D**) Functional annotation of downregulated (red) and upregulated (blue) genes after ZFHX3-KO using the GO::BP (biological processes) terms by DAVID.

## Loss of Zfhx3 affects rhythmic transcription of TTFL and clock-controlled genes

The present and recent studies have demonstrated severe disturbances of daily and circadian behaviour in mice carrying a missense mutation (*Zfhx3*<sup>Sci</sup>) (*Parsons et al., 2015*) or knock-out (*Zfhx3*<sup>Flox/Flox</sup>;UBC-Cre<sup>+</sup>) (*Wilcox et al., 2017*) of ZFHX3. In addition to assessing general gene up- or downregulation in mutants (above), we next wanted to assess how deletion of ZFHX3 might alter transcriptional rhythms. For this, we investigated gene expression levels in the SCN at six distinct times-of-day (*Hughes et al., 2017*), starting at ZT02, at 4 h intervals ('Materials and methods'). In total, we analysed the daily transcriptional profile of 20,521 genes in the SCN and compared the rhythmicity index between the control (Cre-neg) and mutant (+Tamoxifen) groups using statistical framework *dryR* (*Weger et al., 2021*). This enabled us to categorize the genes into one of five distinct modules ('Materials and methods'): module 1 refers to genes with no detected 24 h rhythm in their expression levels in either condition (n = 16,832, *Figure 4—figure supplement 1A*); module 2 consists of genes that were rhythmic in control SCN but lost rhythmicity in the ZFHX3-KO (n = 1566, *Figure 4A*); module 3 consists of genes that gained daily rhythmicity after ZFHX3-KO (n = 403, *Figure 4—figure supplement 1B*); module 4 was composed of rhythmic genes that exhibited no change between control and mutants (n = 1653, *Figure 4B*); and module 5 included genes that retained rhythmic expression but with a change in either amplitude or phase (peak expression) in the mutant group (n = 67) (*Supplementary file 4*, *Figure 5*).

Thus, we noted ~82% of compared genes did not show a daily rhythm (module 1). Conversely, ~18% of gene expressed in the SCN exhibited daily rhythmicity under one or other condition. Of these, ~45% were not affected by loss of ZFHX3, whereas ~55% of rhythmic transcripts showed alteration in their daily pattern following loss of ZFHX3. Indeed, ~42% of rhythmic genes required ZFHX3 to maintain that rhythm, whereas ~2% of genes (403/20,521) gained rhythmicity in the absence of ZFHX3. Thus, ZFHX3 contributes, directly or indirectly, to a significant part of the daily rhythmic transcriptome of the SCN.

For each module, we compiled the gene list and examined the functional enrichment pathways. The genes that lost rhythmic transcription after the deletion of ZFHX3 (module 2) were seen to be involved in intracellular protein transport, signal transduction such as WNT-pathway linking circadian and cell-cycle processes (*Matsu-Ura et al., 2018*), oxidative stress response, etc. (*Figure 4—figure supplement 2A*). On the contrary, genes that gain rhythmicity (module 3) were associated with RNA processing, stress, and metabolic pathways. The genes that did not show any change in rhythmic expression (module 4) were found to be enriched for functional processes such as protein phosphorylation and de-phosphorylation, cytoskeleton organization, DNA repair, etc. Interestingly, only a tiny proportion (~1.8%) of the rhythmic genes showed substantial changes in their daily cyclic gene expression profiles (phase and amplitude, module 5), but these were primarily involved in circadian rhythm generation, entrainment, and CCG expression (*Figure 4—figure supplement 2B*). This suggests that in the absence of ZFHX3 the core circadian mechanism of the SCN is disrupted but is not ablated.

Indeed, almost all the genes driving the cell-autonomous TTFL were severely affected by *Zfhx3* perturbation, with *Bmal1* demonstrating a complete loss of 24 h rhythm (*Figure 4A*), and its counterpart *Clock* mRNA showing overall reduced expression levels (*Supplementary file 3*). PER family genes (*Per1-3*) showed a significant advancement in peak expression with low amplitude (*Figure 5C*). Specifically, we observed a phase advance of ca. 2 h for *Per2* and *Per3*, with *Per1* (*Kuhlman et al., 2003*), which is also known to mediate light-responsive pathways, advancing by only ~0.3 h. Strikingly, there was no obvious change in the rhythmic levels or phase for CRY family genes (*Cry1, 2*) (*Figure 4B*), indicating that ZFHX3 acts only on select genes driving the cell-autonomous clock in the SCN. Given that *Bmal1* transcription is under the control of ROR (*Rora, b, c*) and REV-ERB (*Rev-erb a, b*) factors (*Liu et al., 2008*), we also noted loss of the rhythm in *Rorb* expression, whilst *Nr1d1* showed a phase advance by almost 1 h (*Table 1*).

Next, we examined the effect of loss of ZFHX3 on rhythmic expression of the CCGs. CCGs are basically genes whose expression is regulated by circadian clock factors (such as CLOCK:BMAL1) that are directly associated with the TTFL. Several CCGs like DBP (D site albumin promoter binding protein) show rhythmic transcription (*Lopez-Molina et al., 1997*), and in turn control expression of a repertoire of genes, generating a cascade of 24 h cycling genes in the SCN (*Herzog et al., 2017*). In our mutant group, *Dbp* mRNA levels also presented an advanced peak (~1.66 h) with reduced amplitude

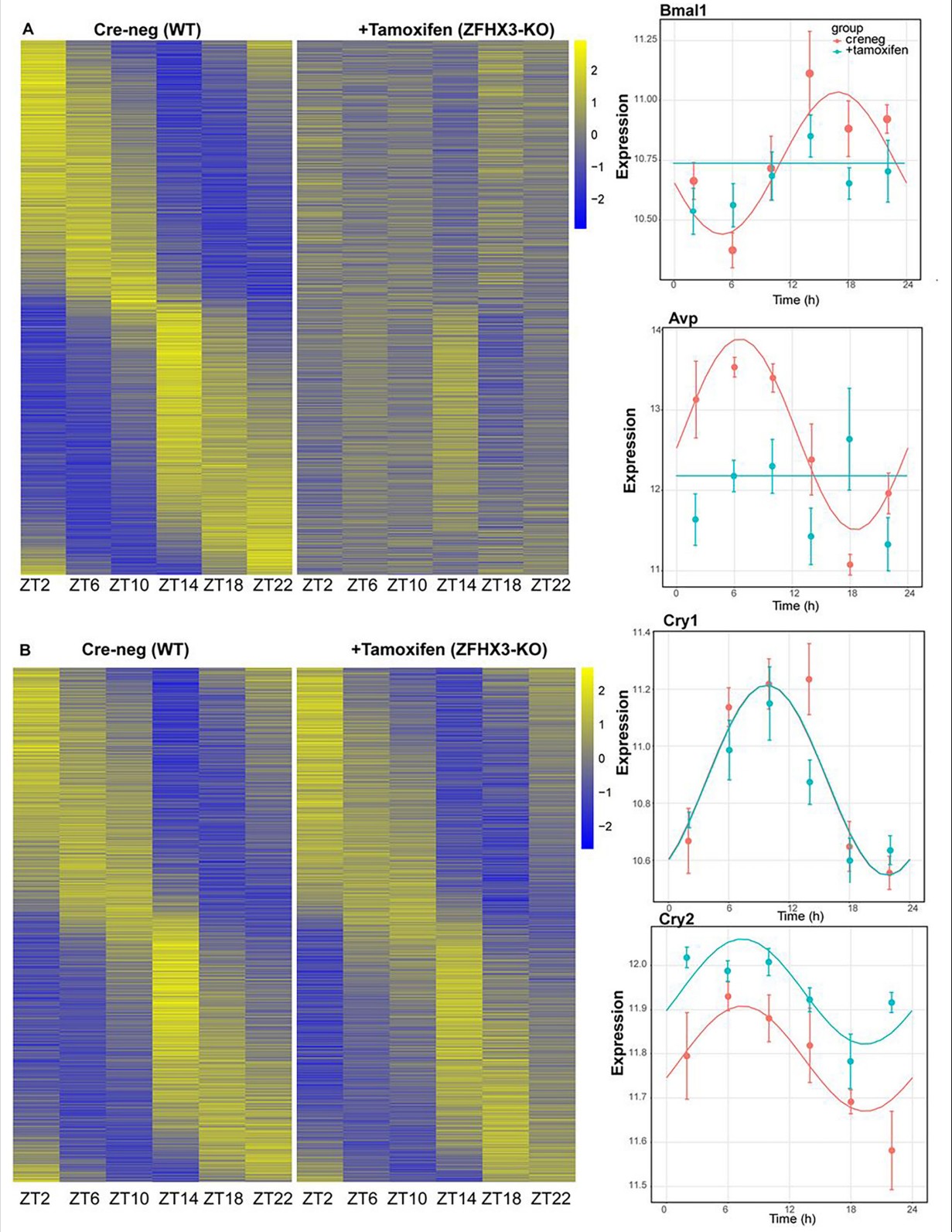

**Figure 4.** Effect of ZFHX3-KO on rhythmic gene expression. (**A**) Left: heatmap showing loss of rhythmic gene expression after *Zfhx3-KO* (module 2) as computed by dryR statistical framework, Right: illustrative examples of daily abundance of module 2 genes, *Bmal1 and Avp,* in Cre-neg and +Tamoxifen conditions. (**B**) Left: heatmap showing no effect on rhythmic gene expression after *Zfhx3-KO* (module 4); right: illustrative examples of module 4 genes, *Cry1 and Cry2,* in Cre-neg and +Tamoxifen conditions.

*Figure 4 continued on next page*

*Figure 4 continued*

The online version of this article includes the following figure supplement(s) for figure 4:

**Figure supplement 1.** Effect of *Zfhx3*-KO on suprachiasmatic nucleus (SCN) transcriptome.

**Figure supplement 2.** Functional annotation of rhythmic genes in the suprachiasmatic nucleus (SCN).

(*Figure 5C*). Another such TF- *Creb3l1* (cAMP-responsive element-binding protein 3-like 1), which has been implicated in intercellular coupling (*Bedont et al., 2017*), showed a distinctive change in phase and amplitude. Therefore, the alteration seen in the rhythmicity index at the level of TTFL genes due to the loss of ZFHX3 was recapitulated in the diurnal expression of certain CCGs in the SCN.

Finally, we studied the effect of ZFHX3 deletion on the rhythmic pattern of intercellular coupling agents (neuropeptides), whose overall mean expression levels were significantly reduced (*Figure 3*, *Vip*, *Grp*). Along with lowered baseline expression, we noted severe loss in 24 h rhythm for crucial SCN neuropeptides such as *Avp* (*Figure 4a*), *Prok2* and *Npy* (*Figure 5c*). For *Avp*, there was no apparent increase in daytime mRNA expression (*Arima et al., 2002*), and the levels remained intermediate throughout the day. Conversely, for both *Prok2* and *Npy*, transcript levels were reduced to minimal levels, possibly resulting in compromised intercellular-coupling in the SCN.

The effects of ZFHX3 on daily transcriptional profiles therefore varied between different gene categories. Whereas rhythmicity of downstream, CCGs was either independent of ZFHX3 or required ZFHX3 to be sustained, core circadian function was sustained but with altered dynamics. Intriguingly, this systematic approach not only highlighted the genes that lost 24 h rhythmic oscillations, such as *Bmal1* and the neuropeptide *Avp* (module 2, *Figure 4A*), but also identified genes that retained rhythmicity with changes in their waveform (module 5). Given the fact that ZFHX3 deficient mice do not lose rhythmic behaviour in a daily LD cycle (*Wilcox et al., 2017*; *Figure 2*), but do exhibit a short circadian period or arrhythmicity in free-running (complete darkness: DD) conditions, we believe our targeted SCN-specific assessment unravels the vital role of ZFHX3 in the mechanistic regulation of the daily molecular clock.

## The state of the daily SCN clock in ZFHX3-deficient mice

In order to link the perturbations of the core molecular clock in the SCN of ZFHX3-deficient mice with their altered behavioural profiles, we used TimeTeller (*Vasilyev, 2025*), a recently established method to predict circadian clock phase alongside the parameter 'Theta' which provides an index of clock disruption from transcriptomic data (*Vlachou et al., 2024*). This approach allows one to assess the clock globally as a system instead of gene by gene. For this, an initial training model was built using the available Cre-neg and -Tamoxifen (n = 35, control) mice data (*Figure 6A*). Cross-validation for the training model suggested that the model is informative and predicts the control SCN time well with a median prediction error of –0.08 h (95% CI –0.83, 1.33) and 0.50 h (95% CI –1.67, 3.00), that is, the difference between the TimeTeller predicted time and the known ZT of SCN collection (*Figure 6—figure supplement 1A*). This was further assessed by predicting SCN timing in an independent dataset of SCN RNA-seq (*Deota et al., 2023*). Similar to the TimeTeller predictions for the control group, median prediction error was 0.00 h (95% CI –2.00, 1.33) for the ad libitum fed and –0.66 h (95% CI –1.83, 1.00) for the temporary night-time-restricted feeding groups. These are not distinguishable from the control SCN in the current study (*Figure 6B*). Similarly, the clock disruption parameter theta is low (medians: 0.03, 0.03, and 0.04) as is expected for a fully functional and well-predictable circadian clock in all three control groups. Furthermore, TT clock functionality parameter theta suggests that +Tamoxifen SCN have a similarly functional daily clock compared to the control SCN (median 0.04 vs. 0.03 in controls; *Figure 6—figure supplement 1B*).

While TT's theta metric suggests the clock in the ZFHX3-deleted SCN is not disrupted as it is indistinguishable from controls, the phase of the +Tamoxifen SCN was consistently advanced by a median 1.83 h (95.0% CI 0.585, 2.67) compared to the current controls and the published data (*Deota et al., 2023*) as illustrated in *Figure 6B*. Of note, detecting this ~2 h phase difference from data sampled at 4 h intervals showcases the advantages in robustness and sensitivity of assessing the clock as a system rather than by a gene-by-gene approach, and TimeTeller as a tool of inquiry. In summary, according to TimeTeller, the SCN daily clock is intact but advanced by approximately 2 h in comparison to the

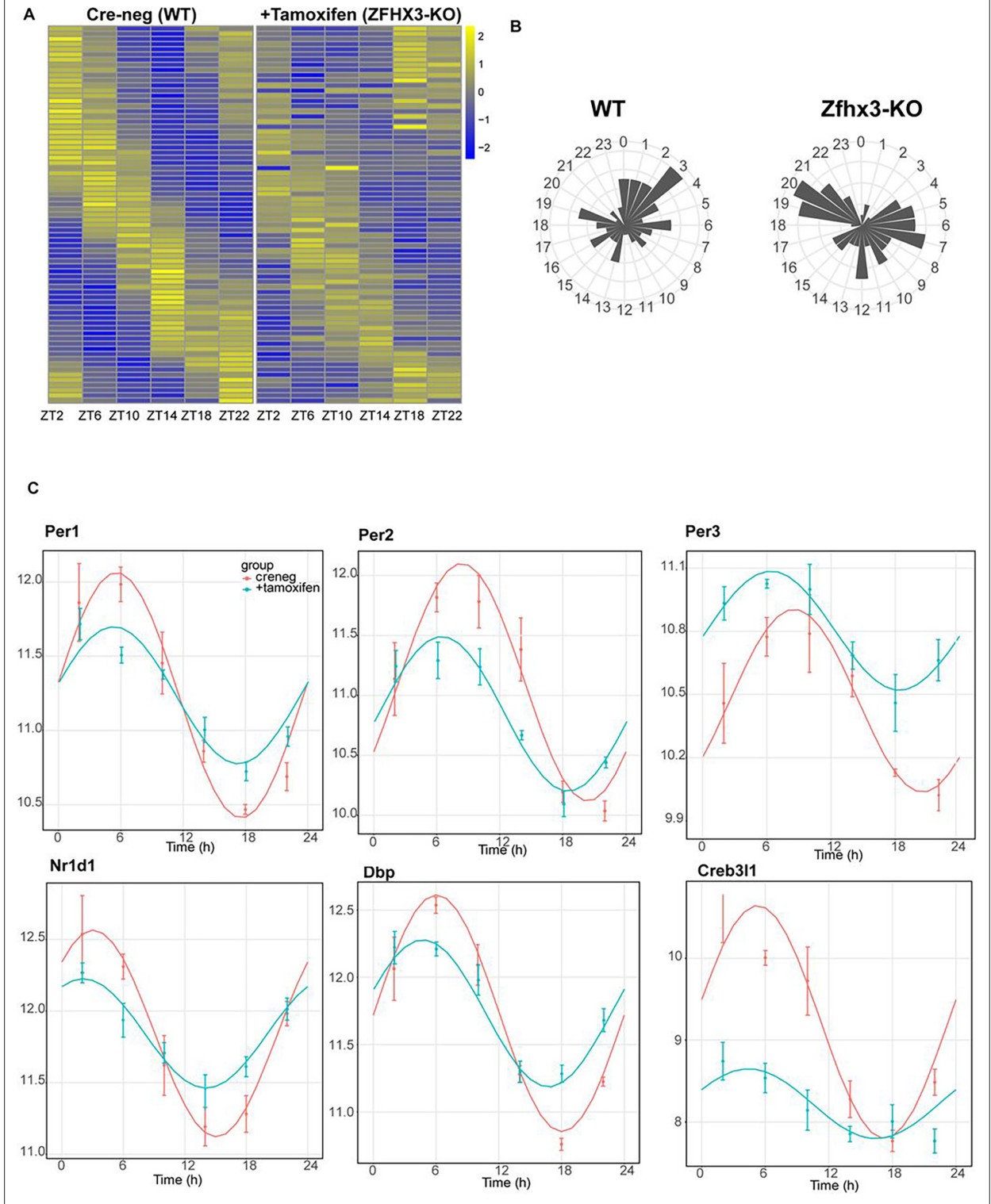

**Figure 5.** Effect of ZFHX3-KO on clock gene expression clustered in module 5. (**A**) Heatmap showing change in rhythmic gene expression after *Zfhx3-KO* (module 5) as computed by dryR statistical framework. (**B**) 24 h radial plots showing the effect on phase (peak expression) of rhythmic genes in Cre-neg (WT) vs. +Tamoxifen (ZFHX3-KO) conditions. (**C**) Illustrative examples of daily abundance of module 5 genes, showing change in phase and amplitude in ZFHX3-KO condition (blue) compared to control (pink). The y-axis values represent normalized expression in the suprachiasmatic nucleus (SCN).

The online version of this article includes the following figure supplement(s) for figure 5:

**Figure supplement 1.** Effect of ZFHX3-KO on circadian clock.

**Table 1.** Effect of loss of ZFHX3 on core-clock genes in the suprachiasmatic nucleus.

| Gene | Rhythm | Phase | Amplitude |
|------|--------|-------|-----------|
| Per1 | Altered | –0.36 | 0.72 (low) |
| Per2 | Altered | –2.0 | 0.69 (low) |
| Per3 | Altered | –2.1 | 0.30 (low) |
| Cry1 | No change | - | - |
| Cry2 | No change | - | - |
| Bmal1 | Lost | - | - |
| Clock | No change | - | - |
| Nr1d1 | Altered | –0.99 | 0.67 (low) |
| Nr1d2 | No change | - | - |
| Rorb | Lost | - | - |

control groups after loss of ZFHX3. This is consistent with the observed daily behavioural activity in mutant mice as shown in *Figure 2*.

## Discussion

The SCN is distinctive in its ability to sustain autonomous rhythmicity which is driven by circadian cascades of transcription that direct metabolic, electrophysiological and signalling rhythms. Our current findings highlight the instrumental role of the TF, ZFHX3, in driving the daily timekeeping mechanism. By comparing the daily behavioural activity of ZFHX3-KO mice with the genetic matched control, we clearly noted an advancement in LD activity and shortening of circadian period in mutants. This finding is consistent with previous studies (*Parsons et al., 2015*; *Wilcox et al., 2021*; *Wilcox et al., 2017*) and prompted a detailed molecular investigation in the SCN. Through targeted ChIP-seq, we successfully demonstrated the genome-wide occupancy of ZFHX3 in SCN chromatin with 78% of ZFHX3 bound sites found to be co-localized with modified histones (H3K4me3, H3K27ac), and a vast majority of these concentrated near gene TSS. This provides strong evidence in support of the observed pervasive regulation conferred by ZFHX3 on the SCN transcriptome. That said, a considerable proportion of ZFHX3 sites were also seen devoid of joint histone modifications (H3K4me3+H3K27ac) and enriched for CTCF binding sites. CTCF is a ubiquitous DNA binding protein that has exceptional roles in genomic organization, chromatin looping, and gene regulation (*Holwerda and de Laat, 2013*; *Kim et al., 2015*; *Mach et al., 2022*). This co-occurrence hints on the potential accessory role of ZFHX3 in modifying genome topology and chromosomal looping to bring about changes in target gene expression by regulating promoter–enhancer interactions (*Pugacheva et al., 2020*). Interestingly, along with CTCF binding sites, ZFHX3 sites were also enriched for other SCN-enriched regulatory factors; RFX, LHX, ZIC (*Araki et al., 2004*; *Hatori et al., 2014*; *Figure 1—figure supplement 1D*), suggesting well-coordinated tissue-specific gene regulation.

Given the crucial role of ZFHX3 in daily timekeeping, we speculated a dynamic change in its genomic occupancy and compared the binding intensity of the TF at two anti-phasic (12 h apart, ZT3 vs. ZT15) timepoints ('Materials and methods'). Surprisingly, we did not detect any differential binding site(s) between the tested timepoints, but rather noticed a uniform and coherent localization. This suggests that ZFHX3 belongs to a family of poised TFs, whose occupancy at regulatory sites does not change during the day, but could potentially act as a co-activator/mediator to modulate oscillating gene transcription. Indeed, in a recent unpublished study (*Del Rocío Pérez Baca et al., 2023*), ZFHX3 was shown to be associated with chromatin remodelling and mRNA processing factors, strengthening the viewpoint as a co-regulator. It is, therefore, highly plausible that ZFHX3 occupies the targeted TSS and partners with clock TFs such as BMAL1 (*CACGTG*), as observed around active sites, to regulate rhythmic gene transcription.

Along with ChIP-seq, we executed detailed RNA sequencing to study the effect of loss of ZFHX3 specifically in the SCN. Crucially, ZFHX3 affected almost a third of the transcriptome in the SCN. An

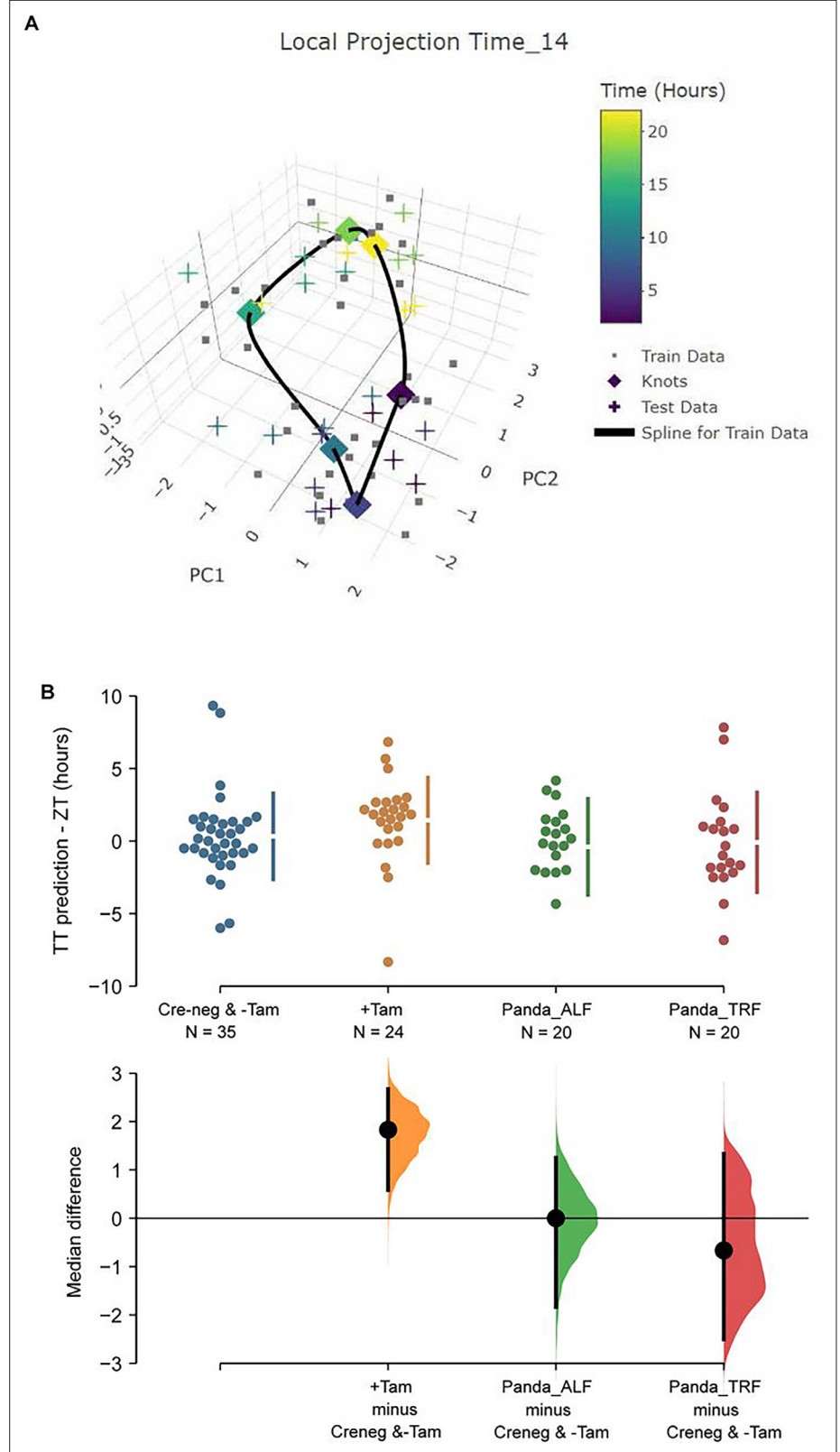

**Figure 6.** TimeTeller model suprachiasmatic nucleus (SCN) circadian clock in control and ZFHX3-KO mice.
(**A**) TimeTeller (TT) model build with the data of all 35 available SCN from Creneg and –Tam mice. Only the knots of the model at the collected ZTs, and predictions for test data of +Tam SCN according to when they were collected are shown. (**B**) Quantitation of SCN phases as difference between ZT to TT prediction for all genotypes

*Figure 6 continued on next page*

*Figure 6 continued*

of this study and previously published wild-type SCNs from ad libitum or night fed mice (Panda_ALF and Panda_TRF). Only the +Tam group is different from Creneg and −Tam control training data (p=0.0028).

The online version of this article includes the following figure supplement(s) for figure 6:

**Figure supplement 1.** Suprachiasmatic nucleus (SCN) circadian clock using TimeTeller.

overall (time-independent) assessment of deletion of *Zfhx3* highlighted a significant impact on the expression of genes involved in several biological processes ranging from cellular differentiation to behavioural responses. We then performed a comprehensive time-dependent assessment to delineate the functional role of ZFHX3 in regulating the circadian timekeeping mechanism. We observed about 55% of rhythmic genes showed alteration at distinct levels in their daily transcriptional pattern following loss of ZFHX3.

At the TTFL level, ZFHX3-KO resulted in the loss of rhythmic *Bmal1* gene expression. It remains to be determined whether this is associated with changes at the post-translational level, given the findings by Abe et al where arrhythmic *Bmal1* transcript did possess posttranslational rhythm (*Abe et al., 2022*), and could explain the subsequent 24 h-driven *Per (1-3)* and *Cry (1-2)* mRNA rhythms noted in our study. Likewise, in a separate study constitutive *Bmal1* expression has been shown to sustain rhythmic levels of transcriptional and translational products of *Per, Cry, Clock, RORc,* and *Nr1d1* by means of mathematic modelling (*Mirsky et al., 2009*). Arguably, clear circadian rhythmicity in BMAL1, CLOCK, or NR1D1 protein levels was not sustained in association with constitutive *Bmal1* levels under in vitro conditions (*Padlom et al., 2022*). Therefore, in future it will be interesting to study the daily abundance of protein levels of the clock genes in *Zfhx3*-deficient SCN, with its associated loss of *Bmal1* mRNA 24 h rhythm, to gain a complete understanding of the clock molecular circuitry.

Broadly speaking, aside from *Bmal1*, the majority of clock genes did not lose 24 h rhythmicity but showed an early peak in expression, ranging between ~0.4 and 2 h that perfectly resonates with the advanced rhythmic behaviour seen in mutants. In this case, it is difficult to delineate a straightforward cause-and-effect axis in terms of the direct targets of the perturbed ZFHX3 activity. Based on the current data, ZFHX3 is possibly partnering with CLOCK:BMAL1 to regulate the daily transcription of clock genes (*Per, Ror, Reverb*) and CCGs. Two possible scenarios could explain the noted effect on the TTFL genes due to lack of ZFHX3 (*Figure 5*). First, ZFHX3 preferentially controls *Per* family gene transcription (and not *Cry*) by binding at the promoter (*Figure 1*). Loss of ZFHX3 affects the daily rhythm of *Per* gene transcription, and potentially PER protein levels, which in turn disturbs the CLOCK:BMAL1-driven transcription cascades. This leads to an aberrant daily transcriptional profile of the *Ror* (activator)/*Reverb* (repressor) genes as observed, which are known to control the amplitude and oscillation of *Bmal1* (*Liu et al., 2008*), and further results in loss of *Bmal1* transcriptional rhythm. Therefore, the observed change in *Ror* and *Reverb* family gene expression levels could be reminiscent of the effect of ZFHX3 on the TTFL. In a second scenario, the constitutive expression of *Bmal1* mRNA could result from the primary effect of ZFHX3 on CLOCK:BMAL1-driven *Ror/Reverb* genes (affecting ROR/REVERB translational products), which does not disrupt the TTFL, but weakens the daily clock. The direct/indirect effect is seen in the cyclic expression of *Per* family genes, with relatively greater phase change in *Per2* and *Per3* than the *Per1* gene (*Kuhlman et al., 2003*; *Figure 5—figure supplement 1*). It is, however, noteworthy that the CRY family of genes (*Cry1, Cry2*) did not show any obvious change in the daily expression and is supposedly shielding the molecular clock, as recently postulated by *Abe et al., 2022*.

As previously reported (*Parsons et al., 2015*), lack of ZFHX3 also resulted in severe reduction in the SCN neuropeptides such as *Avp, Vip, Grp, Prok2*, which potentially compromises intercellular coupling and synchronization of circadian timekeeping. This universal effect, where loss of ZFHX3 does not deter a particular but all the key SCN neuropeptides, clearly points that (1) ZFHX3 is prevalent in almost all neuronal populations within the SCN, which are largely clustered based on their neuropeptide expression (*Morris et al., 2021*; *Wen et al., 2020*); and (2) potentially results in loosely coupled intercellular clocks that are still able to drive 24 h behavioural and molecular rhythms but with advanced phase. The prevalence of ZFHX3 in neuronal populations is also supported by a complete independent assessment in which almost 60% of the top 100 categorized neuronal genes in the SCN (as per single-cell atlas) were seen to be affected by ZFHX3 deletion based on the current dataset. In contrast, only 20% of the 100 most expressed genes in the astrocytic cell populations were impacted.

Along with this, lack of *Zfhx3* also leads to aberrant daily transcriptional rhythms of several CCGs, with the majority of them seen to lose daily oscillations. This could potentially result in hindrance of clock-regulated vital biological processes such as regulation of cell-cycle and metabolic pathways. Indeed, ZFHX3 is reported to be crucial for the development of a functional SCN (*Wilcox et al., 2021*) and could act as an 'integrator' of afferent signals such as light and feeding to generate timely and robust physiological outputs. Moreover, genetic ablation of *Zfhx3* has been shown to result in a compromised light sensing mechanism in retina (*Hughes et al., 2021*), while the missense mutation (*Zfxh3*<sup>Sci/+</sup>) results in reduced growth and altered metabolic gene expression in the arcuate nucleus of the hypothalamus (*Nolan et al., 2023*) converging on its functional roles in other brain regions. The pleiotropic roles of ZFHX3 further extend in non-brain tissues with its recently described crucial involvement in angiogenesis for tumour development (*Fu et al., 2020*), cardiac function (*Jameson et al., 2023*), and spinocerebellar ataxia (*Figueroa et al., 2024*; *Wallenius et al., 2024*). It is important to note that although in the present study we used adult-specific *Zfhx3* null mutants resulting in global loss of ZFHX3, the circadian effects observed both at molecular and behavioural levels are independent of its pleiotropic role(s) in other tissues. In the future, however, it will be beneficial to carry out the investigation using an inducible, SCN-specific genetically ablated model to circumvent any confounding effects. Regardless, the current research is of great importance as it not only maps the genomic binding of ZFHX3, but also demonstrates its critical association with both lineage- and clock-controlling TFs that perfectly resonates with the observed profound effects of ZFHX3 on the SCN transcriptome, regulating a vast majority of genes ranging from circadian timing to synaptic transmission and inflammatory responses. This highlights how ZFHX3 regulates the core-clock (TTFL) in addition to its control over CCGs in order to drive sustained transcriptional rhythms in the SCN. Given the extensive pleiotropy of ZFHX3, it provides a solid substrate to further understand the functional interaction of ZFHX3 in distinct tissues indispensable for overall health and fitness.

## Materials and methods

**Key resources table**

| Reagent type (species) or resource | Designation | Source or reference | Identifiers | Additional information |
|---|---|---|---|---|
| Strain, strain background (*Mus musculus*) | C57BL/6J | Jackson Laboratories | RRID:IMSR_JAX:000664 | Maintained at MRC, Harwell |
| Strain, strain background (*M. musculus*) | UBC-cre (B6.Cg-Tg(UBC-cre/ERT2)1Ejb/J) | Jackson Laboratories | RRID:IMSR_JAX:007001 | |
| Strain, strain background (*M. musculus*) | Zfhx3<sup>Flox</sup> backcrossed to C57BL/6J | *Wilcox et al., 2017* | | MRC, Harwell |
| Antibody | Anti-ZFHX3 (rabbit polyclonal) | *Parsons et al., 2015* | | 12 µl |
| Software, algorithm | ClockLab | https://actimetrics.com/products/clocklab/clocklab-analysis-version-6/ | RRID:SCR_014309 | |
| Software, algorithm | MACS | https://taoliu.github.io/MACS/ | RRID:SCR_013291 | v2.1.0 |
| Software, algorithm | Diffbind | http://bioconductor.org/packages/release/bioc/vignettes/DiffBind/inst/doc/DiffBind.pdf | RRID:SCR_012918 | v2.10.0 |
| Commercial assay or kit | RNeasy Micro Kit | Qiagen | QIAGEN, Cat 74004 | |

*Continued on next page*

*Continued*

| Reagent type (species) or resource | Designation | Source or reference | Identifiers | Additional information |
|---|---|---|---|---|
| Commercial assay or kit | PrecisionPLUS OneStep RT-qPCR Master Mix with ROX | PrimerDesign | https://www.primerdesign.co.uk/media/downloads/df6ea8f4-e857-11ee-8882-fa163e309ced.pdf | |
| Chemical compound, drug | Tamoxifen | Cambridge Bioscience | CAY13258-1g | |
| Software, algorithm | dryR | *Weger et al., 2021* | https://github.com/naef-lab/dryR | |
| Software, algorithm | TimeTeller | *Vlachou et al., 2024* | https://github.com/VadimVasilyev1994/TimeTeller | |

## Mice

All animal studies were performed under the guidance issued by Medical Research Council in Responsibility in the Use of Animals for Medical Research (July1993) and Home Office Project Licence 19/0004. WT C57BL/6J were maintained and provided in-house by MRC Harwell. Animals were group housed (4–5 mice per age) in individually ventilated cages under 12:12 h light-dark conditions with food and water available ad libitum. Adult-specific knock-out *Zfhx3* mice were generated using an inducible Cre line as described by *Wilcox et al., 2017*. Briefly, tamoxifen-inducible Cre mice (UBC-Cre) were crossed to *Zfhx3* floxed (*Zfhx3*$^{Flox}$) mice to produce an initial stock of *Zfhx3*$^{Flox/+}$;UBC-Cre$^+$ mice. These were subsequently crossed to *Zfhx3*$^{Flox/+}$ mice to generate experimental cohorts (*Sun et al., 2012*). Out of the six possible genotype combinations, two genotypes – *Zfhx3*$^{Flox/Flox}$;UBC-Cre$^-$ (mice homozygous for *Zfhx3*$^{Flox}$ allele but not carrying the Cre allele) and *Zfhx3*$^{Flox/Flox}$;UBC-Cre$^+$ (mice homozygous for *Zfhx3*$^{Flox}$ allele and hemizygous for Cre allele) – were assigned as 'control' (Cre-neg) and 'experimental' (+Tamoxifen) groups, respectively, and dosed with tamoxifen. In addition, we also included a cohort of *Zfhx3*$^{Flox/Flox}$;UBC-Cre$^+$ mice without tamoxifen treatment as a secondary control (-Tamoxifen). Animals were group housed (4–5 mice per age) in individually ventilated cages under 12:12 h light-dark conditions with food and water available ad libitum.

## Experimental design

WT C57BL/6J mice (aged between 8 and 12 weeks) were used for SCN tissue punch collection as described by *Jagannath et al., 2013* at ZT3 and ZT15, where lights on at 7 am (ZT0) and lights off at 7 pm (ZT12), for ZFHX3-ChiP-seq. For SCN RNA-seq, mice from control (Cre-neg and -Tamoxifen) and *Zfhx3* knock out (+Tamoxifen) cohorts were used for SCN tissue collection (*Bafna et al., 2023b*) at six distinct timepoints starting from ZT2 at every 4 h. To achieve high-resolution transcriptomic dataset, four biological replicates per timepoint were collected independently from the control (Cre-neg) and experimental (+Tamoxifen) groups. For the second control group, viz., -Tamoxifen, 2–4 biological replicates were collected due to time constraints. Furthermore, each biological replicate for all the three tested condition constituted three individual SCN samples.

## Animal behaviour

Circadian wheel running was performed as previously described (*Banks and Nolan, 2011*). Briefly, mice were singly housed in cages containing running wheels, placed in light-controlled chambers and wheel running activity monitored via ClockLab (Actimetrics). Animals were given a 5-day acclimatization period in a 12 h light/dark cycle (100 lux light intensity) before their activity was monitored for 32 days in constant darkness and 24 days under a 12 h light/dark cycle (100 lux light intensity). Circadian activity analysis was performed upon the entire of the period in constant darkness and upon the final 7 days in the light/dark cycle. Circadian analysis was performed using ClockLab Version 6 (https://actimetrics.com/products/clocklab/clocklab-analysis-version-6/). *Zfhx3*$^{Flox/Flox}$;UBC-Cre$^+$ animals were used for this analysis, with animals being randomly assigned into tamoxifen un-dosed (-Tamoxifen) or tamoxifen dosed (+Tamoxifen) cohorts. The numbers of animals per cohort were: male -Tamoxifen = 8; male +Tamoxifen = 8; female -Tamoxifen = 8; female + Tamoxifen = 8. Statistical analysis and graphing of circadian data were performed using GraphPad Prism 8.

## SCN tissue collection for RNA-seq

Animals were sacrificed at the aforementioned timepoints by cervical dislocation, and brains were removed and flash-frozen on dry ice. The frozen brains were placed on dry ice along with the brain matrix (Kent Scientific, Torrington, CT) and razor blades to chill. The brain dissection was performed on a vibrating microtome (7000smz-2 Vibrotome, Campden Instruments) under chilled conditions (*Bafna et al., 2023b*). First, the frozen brain was placed on the chilled brain matrix and a ca. 3-mm-thick region was removed from the caudal end using a razor blade to discard the cerebellum and attain flat surface. Then, the dissected brain was super-glued to a chilled (2–3°C) metal chuck and placed into the metal specimen tray of the Vibratome. The metal specimen tray was filled with pre-chilled 1X RNase free phosphate-buffered saline (Thermo Fisher Scientific, #AM9625). Coronal slices measuring 250 uM in thickness were cut from rostral to caudal at slow speed (0.07 mm/s) and pre-tested specifications; frequency: 70 Hz; amplitude: 1.00 mm. Once cut, the slice with intact optic chiasm, third ventricle and bilateral SCN tissues was selected and transferred on to pre-chilled glass slides (Thermo Fisher Scientific SuperFrost Plus Adhesion slides) with the help of paint brush. The glass slide was placed on top of cold block, and SCN tissues were collected by pre-chilled forceps and dissection needles under dissection microscope (Nikon SMZ645 Stereo Microscope).

## ZFHX3 ChIP and sequencing

ChIP was conducted using the Active Motif ChIP-IT enzymatic kit (https://www.activemotif.com/catalog/868/chip-it-high-sensitivity) according to the manufacturer's instructions. Briefly, SCN punches were pooled from 80 mice at each designated times (ZT3, ZT15) corresponding to one biological replicate per timepoint and fixed for 15 min at room temperature with gentle shaking in 1% form-aldehyde. The crosslinking reaction was stopped by addition of glycine to a final concentration of 0.125 M. Crosslinked chromatin was sheared with Active Motif's EpiShear probe sonicator (Cat# 53051). Genomic DNA (Input) was prepared by treating aliquots of chromatin with RNase, proteinase K, and heat for de-crosslinking, followed by SPRI beads clean up (Beckman Coulter) and quantitation with Clariostar (BMG Labtech).

24 ug sheared chromatin sample collected from each timepoint (ZT3, ZT15) was incubated with 12 ul of rabbit polyclonal antibody against ZFHX3 (*Parsons et al., 2015*), together with magnetic beads, on a rotator at 4°C overnight. This was followed by crosslinking reversal and proteinase-K digestion as per the manufacturer's protocol. The recovered DNA was purified using PCR purification reagents and subjected to quantitative PCR for quality inspection. ZFHX3 immuno-precipitated genomic DNA along with their corresponding input samples was sequenced on Illumina (75 bp, single end reads: ~30 million reads/sample) NextSeq 500 sequencing platform by Active Motif Epigenomic Services.

## ZFHX3 ChIP-seq peak calling and data analysis

Raw sequence data was quality assessed by removing low-quality bases (Phred < 20) and trimmed using FastQC (https://www.bioinformatics.babraham.ac.uk/projects/fastqc/) and Trimmomatic v0.36 (http://www.usadellab.org/cms/index.php?page=trimmomatic), respectively. FASTQ files containing trimmed sequences were then aligned to mm10 genome assembly to generate binary alignment map (BAM) files and used for peak calling by MACS algorithm v2.1.0 (*Zhang et al., 2008*) at a threshold of q < 0.01. Finally, peaks from ZT3 and ZT15 were analysed for differential binding by a Bioconductor package Diffbind v2.10.0 (http://bioconductor.org/packages/release/bioc/vignettes/DiffBind/inst/doc/DiffBind.pdf) at default settings with p<0.05 and fold change >±2. No significant differential ZFHX3 bound peaks were detected between ZT3 and ZT15. In order to investigate the ZFHX3 peaks overlapping with histone modifications (H3K4me3, H3K27ac) in the SCN, data was downloaded from the National Center for Biotechnology Information (NCBI) Gene Expression Omnibus (GEO; https://www.ncbi.nlm.nih.gov/geo/) accession number GSE217943. The data was normalized using bamCoverage tools, and bigwig files were generated for visual inspection using UCSC Genome Browser and processed using BEDTools suite (*Quinlan and Hall, 2010*). ChIPSeeker package (*Yu et al., 2015*) was used to retrieve the nearest gene around resultant peaks and annotate the genomic region of the peak.

## RNA extraction

Total RNA from each sample replicate was extracted using RNeasy Micro Kit (QIAGEN, #74004) following the protocol for microdissected cryosections as per the manufacturer's guidelines and stored at –80°C. Quality and quantity of RNA were measured using an Agilent Bioanalyzer (Pico chip) and a Nanodrop1000 (Thermo Fisher Scientific, Waltham, MA), respectively.

## RNA-sequencing and data analysis

PolyA RNA-Seq libraries were prepared by Oxford Genomics Centre, University of Oxford, using NEBNext Ultra II Directional RNA Library Prep Kit (NEB, #E7760) and sequenced on NovaSeq6000_150PE (150 bp paired-end directional reads: ~40 million reads/sample) platform. Paired-end FASTQ files were quality assessed (Phred < 20 removed) with FastQC, and Illumina adapters were trimmed with TrimGalore (v0.4.3). Then, the reads were aligned to mm10 genome assembly using STAR (v2.7.8a) with MAPQ value for unique mappers set to 60. BAM files were used to generate read counts per gene by FeatureCounts via Samtools (v1.11). Finally, limma-voom method (*Liu et al., 2015*) from the Bioconductor package-limma (v3.48.0) was adopted to quantify differential gene expression and normalized logarithmic counts per million (CPM) values were generated for downstream analysis. Genes without more than 0.5 CPM in at least two samples are insignificant and filtered out. 8582 of 24,421 (35.14%) genes were filtered out for low expression. TMM was the method used to normalize library sizes.

To study the overlap between genes with ZFHX3 occupancy at TSS and those impacted by *Zfhx3* deletion, R-package GeneOverlap (https://bioconductor.org/packages/release/bioc/html/GeneOverlap.html) was adopted where overlap between two gene lists was tested sing Fisher's exact test with p-value=0, meaning the overlap is highly significant. Fisher's exact test also gives an OR, which represents the strength of association. If an OR is equal to or less than 1, there is no association between the two lists. If the OR is much larger than 1, then the association is strong. The class also calculates the Jaccard index, which measures the similarity between two lists. The Jaccard index varies between 0 and 1, with 0 meaning there is no similarity between the two and 1 meaning the two are identical.

## Differential rhythm analysis

The read counts per gene (n=24,421) for each analysed sample were obtained by FeatureCounts and arranged in sample matrix using coreutils (v8.25). The matrix file was used as an input for *dryR* (R package for Differential Rhythmicity Analysis) available at https://github.com/naef-lab/dryR (*Weger, 2024*) Normalized gene expression from 'control' (Cre-neg) and 'experimental' (+Tamoxifen) groups were categorized into five modules with a threshold of BICW (Bayes information criterion weight) > 0.4 based on their time-resolved expression pattern.

## RT-qPCR

Total RNA was extracted from the SCN tissue as described above from three/four biological replicates per genotype (-Tamoxifen vs. +Tamoxifen), where each biological replicate constitute one independent SCN tissue. 10 ng RNA per sample replicate was tested by one-step RT_qPCR reagent (PrecisionPLUS OneStep RT-qPCR Master Mix with ROX at a reduced level premixed with SYBRgreen, PrimerDesign) using an Applied Biosystems 7500 real-time PCR instrument (10 min at 55°C, 8 min at 95°C, 40 × 10 s at 95°C, and 60 s at 60°C, followed by meltcurve analysis 65–95°C in 0.5°C increments at 5 s per step) in triplicates for *Zfhx3* expression in control and mutant groups. 7500 Applied Biosystems Software v2.3 (https://www.thermofisher.com/uk/en/home/technical-resources/software-downloads/applied-biosystems-7500-real-time-pcr-system.html) was used to obtain the relative quantification values and examine melt curves. All RT-qPCR data were normalized to the *Rpl32* reference control gene and analysed using the standard curve method (*Larionov et al., 2005*). The primers used for *Zfhx3* (CTGTGCCAAGACATGCTCAAC [forward] and CAGCAGGGAGGAACATGCTA [reverse]) and *Rpl32* (AGGCACCAGTCAGACCGATA [forward] and TGTTGGGCATCAGGATCTGG [reverse]) were designed and screened for target specificity with the National Center for Biotechnology Information's (NCBI) Basic Local Alignment Search Tool (BLAST) (https://blast.ncbi.nlm.nih.gov/). The primer sets were validated before use (requirements: no predicted off-targets, primer efficiency 85–120%, no significant signal in nontemplate control, single peak in melt curve, and single band at the predicted

size when separated via agarose gel electrophoresis). ZFHX3 gene expression differences from qPCR were plotted and analysed with GraphPad Prism v9.5.1 for Windows, GraphPad Software (https://www.graphpad.com/). Unpaired-two tailed *t*-test between the control and mutant groups across the tested time points was set at alpha = 0.05.

## Gene annotation

The gene list derived from the Bioconductor-based package ChIPseeker v1.28.3 (*Yu et al., 2015*) was fed into the Database for Annotation, Visualization, and Integrated Discovery (DAVID) tool (*Dennis et al., 2003*). The functional annotation chart based on KEGG pathway, Gene Ontology (GO): biological processes (BP) was plotted with the help of the ggplot2 package in Rv4.0.5 (https://ggplot2.tidyverse.org/).

## HOMER motif analysis

The findMotifsGenome.pl function within the HOMER v4.11 package (*Heinz et al., 2010*) was used to identify enriched motifs and their corresponding TFs with options size 200 –len 8, 10, 12 –mask –preparse –dumpfasta, with default background regions.

## TimeTeller modelling of the SCN clock

TimeTeller (TT) analysis was conducted using all 35 available SCN time-course RNA-seq samples from control (Cre-neg and -Tamoxifen) mice to train the model (*Vlachou et al., 2024*). Data from +Tamoxifen SCN were then projected into the model as test samples. Based on the available training data, we selected the 10 most rhythmic and consistent genes to build the TimeTeller model, in this case: *Bmal1, Per2, Cry1, Tef, Hlf, Dbp, Nr1d2, Npas2, Ciart,* and *Bhlhe41*. As a first step to cross-validate the model, we used a 'leave-one-out' approach. Iteratively, 5–6 randomly chosen samples from all Cre-neg and -Tamoxifen SCN were left out from the training data to build the TT model, and the left-out samples were treated as test samples to determine the TT prediction for all Cre-neg and -Tamoxifen SCN. In addition, we used published SCN RNA-seq data to independently validate the full model with all 35 control SCN from this study (*Deota et al., 2023*). TT analysis predicts phase and clock functionality (theta) based on the averaged likelihood for the projected test sample. For the SCN model, we used time-course normalization with log threshold selected at –6. Further, to compare TT phase predictions between genotypes, we used estimation statistics (*Ho et al., 2019*) and p-values were generated using a permutation testing with 5000 bootstrap samples.

# Acknowledgements

We thank the staff of the Mary Lyon Centre and core services at MRC Harwell Institute for assistance with mouse studies. We thank Dr Benjamin Weger, University of Queensland, for his valuable suggestions and timely inputs during data analysis using dryR. We also thank Michelle Simon and Richard Reeves for ChIP-seq data processing and accessibility on the UCSC Genome Browser. PMN and GB were supported by the Medical Research Council (MC_U142684173). AB was supported by the Medical Research Council (MC_U142684173) and Biotechnology and Biological Sciences Research Council (BB/Z514792/1). MHH was supported by the Medical Research Council (MC_U105170643). RD was supported by the Cancer Research UK and EPSRC (C53561/A19933), and VV was funded by the UK Medical Research Council Doctoral Training Partnership (MR/N014294/1).

# Additional information

### Funding

| Funder | Grant reference number | Author |
| --- | --- | --- |
| Medical Research Council | MC_U142684173 | Akanksha Bafna<br>Gareth Banks<br>Patrick M Nolan |

| Funder | Grant reference number | Author |
| --- | --- | --- |
| Biotechnology and Biological Sciences Research Council | BB/Z514792/1 | Akanksha Bafna |
| Medical Research Council | MC_U105170643 | Michael H Hastings |
| Engineering and Physical Sciences Research Council | C53561/A19933 | Robert Dallmann |
| Medical Research Council | MR/N014294/1 | Vadim Vasilyev |

The funders had no role in study design, data collection and interpretation, or the decision to submit the work for publication.

### Author contributions

Akanksha Bafna, Resources, Formal analysis, Validation, Investigation, Methodology, Writing – original draft; Gareth Banks, Conceptualization, Investigation, Visualization, Methodology, Writing – review and editing; Vadim Vasilyev, Validation, Writing – review and editing; Robert Dallmann, Software, Investigation, Writing – review and editing; Michael H Hastings, Conceptualization, Supervision, Visualization, Writing – review and editing; Patrick M Nolan, Conceptualization, Resources, Supervision, Funding acquisition, Writing – original draft, Project administration, Writing – review and editing

### Author ORCIDs

Akanksha Bafna https://orcid.org/0000-0002-6666-5318
Robert Dallmann https://orcid.org/0000-0002-7490-0218
Michael H Hastings https://orcid.org/0000-0001-8576-6651

### Ethics

All animal studies were performed under the guidance issued by Medical Research Council in Responsibility in the Use of Animals for Medical Research (July1993) and Home Office Project Licence 19/0004.

Reviewer #1 (Public review): https://doi.org/10.7554/eLife.102019.3.sa1
Reviewer #2 (Public review): https://doi.org/10.7554/eLife.102019.3.sa2
Author response https://doi.org/10.7554/eLife.102019.3.sa3

## Additional files

### Supplementary files

Supplementary file 1. List of ZFHX3-binding sites in the SCN.

Supplementary file 2. Circadian wheel running analysis comparing $Zfhx3^{Flox/Flox}$;UBC animals dosed with Tamoxifen (+Tamoxifen) to those without Tamoxifen dosing (-Tamoxifen).

Supplementary file 3. Gene expression changes after the loss of ZFHX3.

Supplementary file 4. Effect of ZFHX3-KO on rhythmic gene expression in the SCN.

MDAR checklist

### Data availability

All raw and processed sequencing data generated in this study have been submitted to the NCBI Gene Expression Omnibus (https://www.ncbi.nlm.nih.gov/geo/query/acc.cgi?acc=GSE261795) under accession number GSE261795 .

The following dataset was generated:

| Author(s) | Year | Dataset title | Dataset URL | Database and Identifier |
| --- | --- | --- | --- | --- |
| Bafna A, Banks G, Hastings MH, Nolan PM | 2025 | ZFHX3 Chipseq and RNASeq | https://www.ncbi.nlm.nih.gov/geo/query/acc.cgi?acc=GSE261795 | NCBI Gene Expression Omnibus, GSE261795 |

The following previously published dataset was used:

| Author(s) | Year | Dataset title | Dataset URL | Database and Identifier |
|---|---|---|---|---|
| Bafna A, Banks G, Hastings MH, Nolan PM | 2023 | SCN Histone ChIPSeq data | https://www.ncbi.nlm.nih.gov/geo/query/acc.cgi?acc=GSE217943 | NCBI Gene Expression Omnibus, GSE217943 |

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
