## [Editor Report · eLife Assessment]

This is an **important** study that generates an inventory of accessible genomic regions bound by a transcription factor ZFHX3 within the suprachiasmatic nucleus in the hypothalamus and details the impact of its depletion on daily rhythms in behaviour and gene expression patterns. Analysis using circadian phase-estimation algorithms makes the argument that gene regulatory networks are at play and changes in gene expression of a few clock genes cannot account for the observed animal behaviour. While the transcriptome analysis is **compelling**, the data on the activity of the TF in rhythmic gene expression is **solid**, and interpretations that allow for direct and/or indirect roles have been incorporated.

---

## [Referee Report · Reviewer #1 (Public review)]

Summary:

Authors of this article have previously shown the involvement of the transcription factor Zinc finger homeobox-3 (ZFHX3) in the function of the circadian clock and the development/differentiation of the central circadian clock in the suprachiasmatic nucleus (SCN) of the hypothalamus. Here, they show that ZFHX3 plays a critical role in the transcriptional regulation of numerous genes in the SCN. Using inducible knockout mice, they further demonstrate that the deletion Of Zfhx3 induces a phase advance of the circadian clock, both at the molecular and behavioral levels.

Strengths:

- Inducible deletion of Zfhx3 in adults

- Behavioral analysis

- Properly designed and analyzed ChIP-Seq and RNA-Seq supporting the conclusion of the behavioral analysis

Comments on revisions:

The authors have properly addressed reviewers' issues.

---

## [Referee Report · Reviewer #2 (Public review)]

Summary

ZFHX3 is a transcription factor expressed in discrete populations of adult SCN and was shown by the authors previously to control circadian behavioral rhythms using either a dominant missense mutation in Zfhx3 or conditional null Zfhx3 mutation using the Ubc-Cre line (Wilcox et al., 2017). In the current manuscript, the authors assess the function of ZFHX3 by using a multi-omics approach including ChIPSeq in wildtype SCNs and RNAseq of SCN tissues from both wildtype and conditional null mice. RNAseq analysis showed a loss of oscillation in Bmal1 and changes in expression levels of other clock output genes. Moreover, a phase advance gene transcriptional profile using the TimeTeller algorithm suggests the presence of a regulatory network that could underlie the observed pattern of advanced activity onset in locomotor behavior in knockout mice.

In Figure 1, the authors identified the ZFHX3 bound sites using ChIPseq and compared the loci with other histone marks that occur at promoters, TSS, enhancers and intergenic regions. And the analysis broadly points to a role for ZFHX3 in transcriptional regulation. The vast majority of nearly 40000 peaks overlapped H3K4me3 and K27ac marks, active promoters which also included genes falling under the GO category circadian rhythms. However, no significant differential ZFHX3 bound peaks were detected between ZT3 and ZT15. In these experiments, it is not clear if and how the different ChIP samples (ZFHX3 and histone PTM ChIPs) were normalized/downsampled for analysis. Moreover, it seems that ZFHX3 binding or recruitment has little to do with whether the promoters are active.

Based on an enrichment of ARNT domains next to K4Me3 and K27ac PTMs, the authors propose a model where the core-clock TFs and ZFHX3 interact. If the authors develop other assays beyond just predictions to test their hypothesis, it would strengthen the argument for a role in circadian transcription in the SCN. It would be important in this context to perform a ChIP-seq experiment for ZFHX3 in the knockout animal (described from Figure 2 onwards) to eliminate the possibility of non-specific enrichment of signal from "open chromatin'. Alternatively, a ChIPseq analysis for BMAL1 or CLOCK could also strengthen this argument to identify the sites co-occupied by ZFHX3 and core-clock TFs.

Next, they compared locomotor activity rhythms in floxed mice with or without tamoxifen treatment. As reported before in Wilcox et al 2017, the loss of ZFHX3 led to a shorter free running period and reduced amplitude and earlier onset of activity. Overall, the behavioral data in Figure 2 and supplementary figure 2 has been reported before and are not novel.

Next, the authors performed RNAseq at 4hr intervals on wildtype and knockout animals maintained in light/dark cycles to determine the impact of loss of ZFHX3. Overall transcriptomic analysis indicated changes in gene expression in nearly 36% of expressed genes, with nearly half being upregulated while an equal fraction was downregulated. Pathways affected included mostly neureopeptide neurotransmitter pathways. Surprisingly, there was no correlation between the direction in change in expression and TF binding since nearly all the sites were bound by ZFHX3 and the active histone PTMs. The ChIP-seq experiment for ZFHX3 in the UBC-Cre+Tam mice again could help resolve the real targets of ZFHX3 and the transcriptional state in knockout animals.

To determine the fraction of rhythmic transcripts, Using dryR, the authors categorise the rhythmic transcriptome (about 7% in all) into modules that include genes that lose rhythmicity in the KO, gain rhythmicity in the KO or remain unaffected or partially affected. The analysis indicates that a large fraction of the rhythmic transcriptome is affected in the KO model. However, among core-clock genes only Bmal1 expression is affected showing a complete loss of rhythm. The authors state a decrease in Clock mRNA expression (line 294) but the panel figure 4A does not show this data. Instead it depicts the loss in Avp expression - {{ misstated in line 321 we noted severe loss in 24-h rhythm for crucial SCN neuropeptides such as Avp (Fig. 3a).}}

However, core-clock genes such as Pers and Crys show minor or no change in expression patterns while Per2 and Per3 show a ~2hr phase advance. While these could only weakly account for the behavioral phase advance, the authors used TimeTeller to assess circadian phase in wildtype and ZFHX3 deficient mice. This approach clearly indicated that while the clock is not disrupted in the knockout animals, the phase advance can be correctly predicted from a network of gene expression patterns.

Strengths

The authors use a multiomic strategy in order to reveal the role of the ZFHX3 transcription factor with a combination of TF and histone PTM ChIPseq, time-resolved RNAseq from wildtype and knockout mice and modeling the transcriptomic data using TimeTeller. The RNAseq experiments are nicely controlled and the analysis of the data indicates a clear impact on gene-expression levels in the knockout mice and the presence of a regulatory network that could underlie the advanced activity onset behavior.

Weaknesses

It is not clear whether ZFHX3 has a direct role in any of the processes and seems to be a general factor that marks H3K4me3 and K27ac marked chromatin. Why it would specifically impact the core-clock TTFL clock gene expression or indeed daily gene expression rhythms is not clear either. Details for treatment of different ChIP samples (ZFHX3 and histone PTM ChIPs) on data normalization for analysis are needed. The loss of complete rhythmicity of Avp and other neuropeptides or indeed other TFs could instead account for the transcriptional deregulation noted in the knockout mice.

Comments on revisions:

The authors addressed the majority of my criticisms. They also explained that some requested experiments are beyond the scope of the current manuscript, while others are technically not feasible. I do not have any further concerns.

---

## [Author Response]

The following is the authors’ response to the original reviews.

**Public Reviews:**

**Reviewer #1 (Public review):**
Summary:Authors of this article have previously shown the involvement of the transcription factor Zinc finger homeobox-3 (ZFHX3) in the function of the circadian clock and the development/differentiation of the central circadian clock in the suprachiasmatic nucleus (SCN) of the hypothalamus. Here, they show that ZFHX3 plays a critical role in the transcriptional regulation of numerous genes in the SCN. Using inducible knockout mice, they further demonstrate that the deletion Of Zfhx3 induces a phase advance of the circadian clock, both at the molecular and behavioral levels.Strengths:- Inducible deletion of Zfhx3 in adults- Behavioral analysis- Properly designed and analyzed ChIP-Seq and RNA-Seq supporting the conclusion of the behavioral analysisWeaknesses:- Further characterization of the disruption of the activity of the SCN is required.

(1) We thank the reviewer for their valuable inputs. Indeed, a comprehensive behavioral assessment of mice of this genotype was executed in Wilcox et al. ;2017 study. In Wilcox et al.; 2017, Figure 4, 6-h phase advance (jetlag) clearly showed faster reentrainment in ZFHX3-KO mice when compared to the controls.

- The description of the controls needs some clarification.

(2) We agree with the reviewer and have modified the text at line 211-212 to clearly describe the controls.

**Reviewer #2 (Public review):**
Summary:ZFHX3 is a transcription factor expressed in discrete populations of adult SCN and was shown by the authors previously to control circadian behavioral rhythms using either a dominant missense mutation in Zfhx3 or conditional null Zfhx3 mutation using the Ubc-Cre line (Wilcox et al., 2017). In the current manuscript, the authors assess the function of ZFHX3 by using a multi-omics approach including ChIPSeq in wildtype SCNs and RNAseq of SCN tissues from both wildtype and conditional null mice. RNAseq analysis showed a loss of oscillation in Bmal1 and changes in expression levels of other clock output genes. Moreover, a phase advance gene transcriptional profile using the TimeTeller algorithm suggests the presence of a regulatory network that could underlie the observed pattern of advanced activity onset in locomotor behavior in knockout mice.In figure1, the authors identified the ZFHX3 bound sites using ChIPseq and compared the loci with other histone marks that occur at promoters, TSS, enhancers and intergenic regions. And the analysis broadly points to a role for ZFHX3 in transcriptional regulation. The vast majority of nearly 40000 peaks overlapped H3K4me3 and K27ac marks, active promoters which also included genes falling under the GO category circadian rhythms. However, no significant differential ZFHX3 bound peaks were detected between ZT3 and ZT15. In these experiments, it is not clear if and how the different ChIP samples (ZFHX3 and histone PTM ChIPs) were normalized/downsampled for analysis. Moreover, it seems that ZFHX3 binding or recruitment has little to do with whether the promoters are active.

(3) We thank the reviewer for their valuable comment. Different ChIP samples (ZFHX3 and histone PTM ChIPs) were treated in the same manner from preprocessing (quality control by FastQC, adapter trimming, alignment to mm10 genome) and peak calling was performed using respective input samples as control using MACS2 as mentioned in Methods. The data was normalized using bamCoverage tools and bigwig files were generated for visual inspection using UCSC Genome Browser. These additional details are added to Methods at line 592. Finally, BEDTools was employed to study overlapping peaks between ZFHX3 and histone PTMs.

We agree that, alone, the current data does not make any claim for ZFHX3 being crucial for promoter to be active. Our data clearly suggests that a vast majority of ZFHX3 genomic binding in the SCN was observed at active promoters marked by H3K4me3 and H3K27ac and potentially regulating gene transcription.

Based on a enrichment of ARNT domains next to K4Me3 and K27ac PTMs, the authors propose a model where the core-clock TFs and ZFHX3 interact. If the authors develop other assays beyond just predictions to test their hypothesis, it would strengthen the argument for role in circadian transcription in the SCN. It would be important in this context to perform a ChIP-seq experiment for ZFHX3 in the knockout animal (described from Figure 2 onwards) to eliminate the possibility of non-specific enrichment of signal from "open chromatin'. Alternatively, a ChIPseq analysis for BMAL1 or CLOCK could also strengthen this argument to identify the sites co-occupied by ZFHX3 and core-clock TFs.

(4a) We agree that follow-up experiments such as BMAL1/CLOCK ChIPseq suggested by the reviewer will further confirm the proposed interaction of ZFHX3 with core-clock TFs. However, this is beyond the scope of the current study.

(4b) Again, conducting complementary ChIPseq in ZFHX3 knockout mice will strengthen the findings, but conducting TF-ChIPseq in a specific brain tissue such as the SCN (unlike peripheral tissues such as liver) does not only warrant use of multiple animals per sample but is also technically challenging and time-consuming to ensure specificity of the sample. For these reasons, datasets such as ours on the SCN are uncommon. Furthermore, in this particular context, we are certain that, based on current dataset, the ZFHX3 peaks (narrow) we observed were well-defined and met the specified statistical criteria mitigating any risk of signal arising from non-specific enrichment from open-chromatin regions.

Next, they compared locomotor activity rhythms in floxed mice with or without tamoxifen treatment. As reported before in Wilcox et al 2017, the loss of ZFHX3 led to a shorter free running period and reduced amplitude and earlier onset of activity. Overall, the behavioral data in Figure 2 and supplementary figure 2 has been reported before and are not novel.

(5) We recognise that a detailed circadian behavior assessment from adult mice lacking ZFHX3 has been conducted previously by Nolan lab (Wilcox et al; 2017). In the current study, however, we used a separate cohort of mice, to focus on the behavioral advance noted in 24-h LD cycle and generated a more refined assessment. Importantly, these mice were also used for transcriptomic studies as detailed in Figure 3, which we consider to be a positive feature of our experimental design: behavior and molecular analyses were performed on the same animals.

Next, the authors performed RNAseq at 4hr intervals on wildtype and knockout animals maintained in light/dark cycles to determine the impact of loss of ZFHX3. Overall transcriptomic analysis indicated changes in gene expression in nearly 36% of expressed genes, with nearly half being upregulated while an equal fraction was downregulated. Pathways affected included mostly neureopeptide neurotransmitter pathways. Surprisingly, there was no correlation between the direction in change in expression and TF binding since nearly all the sites were bound by ZFHX3 and the active histone PTMs. The ChIP-seq experiment for ZFHX3 in the UBC-Cre+Tam mice again could help resolve the real targets of ZFHX3 and the transcriptional state in knockout animals.

(6) We agree with the reviewer that most of the differentially expressed genes showed ZFHX3 binding at active promoter sites. That said, the current dataset is in line with recently published ZFHX3-CHIPseq data by Baca et al; 2024 [PMID: 38412861] in human neural stem cells and Hu et al; 2024 [PMID: 38871709] in human prostate cancer cells that clearly suggests ZFHX3 binds at active promoters and act as chromatin remodellers/mediators that modulate gene transcription depending on the accessory TFs assembled at target genes. Therefore, finding no correlation in the direction of change in expression is not striking.

To determine the fraction of rhythmic transcripts, Using dryR, the authors categorise the rhythmic transcriptome into modules that include genes that lose rhythmicity in the KO, gain rhythmicity in the KO or remain unaffected or partially affected. The analysis indicates that a large fraction of the rhythmic transcriptome is affected in the KO model. However, among core-clock genes only Bmal1 expression is affected showing a complete loss of rhythm. The authors state a decrease in Clock mRNA expression (line 294) but the panel figure 4A does not show this data. Instead it depicts the loss in Avp expression - {{ misstated in line 321 we noted severe loss in 24-h rhythm for crucial SCN neuropeptides such as Avp (Fig. 3a).}}

(7a) Indeed, among the core-clock genes rhythmic expression is lost after ZFHX3 knockout only for Bmal1. However, given the mice were rhythmic (as assessed by wheel-running activity) in LD conditions, the observed 24-h gene expression rhythm in the majority of core-clock genes (Pers and Crys) is consistent with behavior data, and suggests towards an altered molecular clock with plausible scenarios as explained at line 439. That said, the unique and well-defined changes (amplitude and phase) observed as demonstrated in Figure 5 highlights a model in which ZFHX3 exerts differential control, for example in case of Per2 noted advance in molecular rhythm (~2-h), but no such change in Cry, presents an opportunity to delineate further the regulation of TTFL genes.

(7b) Line 294 revised as – “Bmal1 demonstrating a complete loss of 24-h rhythm (Fig. 4A), and its counterpart Clock mRNA showing overall reduced expression levels (Supplementary Table 3)”.

(7c) Line 321 is referring to loss of Avp expression and the typo has been corrected from “Figure 3a to 4a”. Thank you.

However, core-clock genes such as Pers and Crys show minor or no change in expression patterns while Per2 and Per3 show a ~2hr phase advance. While these could only weakly account for the behavioral phase advance, the authors used TimeTeller to assess circadian phase in wildtype and ZFHX3 deficient mice. This approach clearly indicated that while the clock is not disrupted in the knockout animals, the phase advance can be correctly predicted from a network of gene expression patterns.Strengths:The authors use a multiomic strategy in order to reveal the role of the ZFHX3 transcription factor with a combination of TF and histone PTM ChIPseq, time-resolved RNAseq from wildtype and knockout mice and modeling the transcriptomic data using TimeTeller. The RNAseq experiments are nicely controlled and the analysis of the data indicates a clear impact on gene-expression levels in the knockout mice and the presence of a regulatory network that could underlie the advanced activity onset behavior.Weaknesses:It is not clear whether ZFHX3 has a direct role in any of the processes and seems to be a general factor that marks H3K4me3 and K27ac marked chromatin. Why it would specifically impact the core-clock TTFL clock gene expression or indeed daily gene expression rhythms is not clear either. Details for treatment of different ChIP samples (ZFHX3 and histone PTM ChIPs) on data normalization for analysis are needed. The loss of complete rhythmicity of Avp and other neuropeptides or indeed other TFs could instead account for the transcriptional deregulation noted in the knockout mice.

(8) We thank the reviewer for the constructive feedback. The current data suggests ZFHX3 acts as a mediating factor, occupying targeted active promoter sites and regulating gene expression by partnering with other key TFs in the SCN. Please see point 6 for clarification. The binding sites of ZFHX3 clearly showed enrichment for E-box(CACGTG) motif bound by CLOCK/BMAL1 along with binding sites for key SCN-specific TFs such as RFX (please see Supplementary Fig1). Our data thereby shows that it affects both core-clock and clock output genes (at varied levels) thereby exercising a pervasive control over the SCN transcriptome.

For treatment of ChIP samples please see point 3. We followed ENCODE guidelines strictly.

**Recommendations for the authors**:
**Reviewer #1 (Recommendations for the authors):**
- The early activity onset associated with a short photoperiod is a phenotype found in mice with a perturbed function of the SCN like Per2 mutant (PMID: 17218255), or Clock KO (PMID: 22431615). Such disruption of the SCN function also leads to a faster synchronization to day feeding (PMID: 23824542) or jetlag (PMID: 25063847; PMID: 24092737). Therefore, authors should study the synchronizing function of these mice to day feeding and/or jetlag.

(9) Please see our response to point 1.

- The description of the negative controls needs clarification. While the "Method" suggests that both Cre- and Cre+ mice are treated with Tamoxifen, the text rather suggest that the controls are Cre- and Cre+ animals non-treated by Tamoxifen. Because of the potential effect of Tamoxifen on gene expression, Cre- treated animals are a required control.

(10) We thank the reviewer. As detailed in Methods, both Cre- and Cre+ mice were treated with Tamoxifen and compared. The text had been revised at line 212. In addition to this, another genetic control (-Tamoxifen) was also used (Figure 2 and 3).

- On line 486, authors wrote "It is important to note that although in the present study we used adult-specific Zfhx3 null mutants resulting in global loss of ZFHX3, the effects observed both at molecular and behavioural levels are independent of its functional role(s) in other tissues." On what evidence is this statement based? Using global KO rather suggest a potential role of other tissues.

(11) We agree with the reviewer, but at line 486 we refer to the effects observed at circadian behavior and daily gene expression in the SCN to be independent of pleiotropic roles of ZFHX3 such as involvement in angiogenesis, spinocerebellar ataxia etc. We have revised the text.

**Reviewer #2 (Recommendations for the authors):**
It is not clear whether the behavioral experiments presented in this study were performed on a new set of animals - different from the cohort used in the Wilcox et al 2017 paper. For example, the proportion of total activity graphed in Figure 2C look strikingly similar to activity counts in Figure 3A in the prior publication (doi: 10.1177/0748730417722631)- down to the small burst in activity after ZT20 in the control (-Tam) group.

(12) The behavioral experiments presented in this study were performed on a completely new cohort of mice to those used in Wilcox et al.; 2017. The mice used for behavioral assessment. In the current study were later used for molecular experiments. Please see point 5.

Information on ChIP-seq such as read length, PE or SE seq, number of reads/replicate/condition/sample is missing. Versions of the softwares used should be indicated if known.

(13) The details are added as:

(13a) “Briefly, SCN punches were pooled from 80 mice at each. designated times (ZT3, ZT15) corresponding to one biological replicate per timepoint” at line 567.

(13b) “24 ug sheared chromatin sample collected from each time point (ZT3, ZT15)” at line 571.

(13c) “75-bp single end sequencing : 30 million reads/sample” at line 577.

(13d) “At line 584 – MACS algorithm v2.1.0 added”

Versions of other softwares used were already mentioned.